# Genetic Evidence Prioritizes Neurocognitive Decline as a Causal Driver of Sleep Disturbances: A Multi-Omics Analysis Identifying Causal Genes and Therapeutic Targets

**DOI:** 10.3390/cimb47110967

**Published:** 2025-11-20

**Authors:** Yanan Du, Xiao-Yong Xia, Zhu Ni, Sha-Sha Fan, Junwen He, Yang He, Xiang-Yu Meng, Xu Wang, Xuan Xu

**Affiliations:** 1School of Life Sciences, Anhui Medical University, Hefei 230032, China; duyanan@ahmu.edu.cn (Y.D.); 17719426997@163.com (X.-Y.X.); zhuni@stu.ahmu.edu.cn (Z.N.); 2School of Health Management, Anhui Medical University, Hefei 230032, China; 3Health Science Center, Medical School, Hubei Minzu University, Enshi 445000, China; fanshasha15@163.com (S.-S.F.); mengxy_whu@163.com (X.-Y.M.); 4College of Informatics, Huazhong Agricultural University, Wuhan 430070, China; hjwing@webmail.hzau.edu.cn (J.H.); he_yang@webmail.hzau.edu.cn (Y.H.)

**Keywords:** Alzheimer’s disease, sleep, integrative genomics, machine learning, mitochondrial dysfunction

## Abstract

To resolve the ambiguous causal relationship between sleep disturbances and neurodegenerative diseases such as Alzheimer’s disease (AD), we conducted a multi-stage genetic and multi-omics investigation. Our large-scale bidirectional Mendelian randomization analysis identified a robust, asymmetrical pattern of genetic association, providing strong genetic evidence suggesting that liability for neurocognitive decline and AD is associated with sleep disturbances, with substantially weaker evidence for the reverse direction. To identify the underlying molecular drivers, a multi-omics Summary-data-based MR (SMR) analysis prioritized high-confidence causal genes, including *YWHAZ*, *NT5C2*, *COX6B1*, and *CDK10*. The predictive power of this gene signature was confirmed using machine learning models (ROC-AUC > 0.8), while functional validation through bulk and single-cell transcriptomics uncovered profound, cell-type-specific dysregulation in the AD brain, most notably opposing expression patterns between neurons and glial cells (e.g., *YWHAZ* was upregulated in excitatory neurons but downregulated in glia). Functional enrichment and network analyses implicated two core pathways—nucleotide metabolism centered on *NT5C2* and synaptic function involving *YWHAZ*—and our investigation culminated in the identification of a promising therapeutic interaction, with molecular docking validating high-affinity binding between Ecdysterone and COX6B1 (docking score = −5.73 kcal/mol). Collectively, our findings strengthen the evidence that sleep disruption as a likely consequence of neurodegenerative processes and prioritize a set of validated, cell-type-specific gene targets within critical pathways, offering promising new avenues for therapeutic development.

## 1. Introduction

Sleep, cognition, and neurodegenerative diseases such as Alzheimer’s disease (AD) are intricately linked, with disruptions in one domain often exacerbating pathologies in others. Chronic sleep disturbances, including insomnia and altered chronotypes, are prevalent in aging populations and have been increasingly recognized as modifiable risk factors for cognitive decline and AD progression. Mechanistically, the sleep–wake cycle is thought to play a crucial role in the clearance of neurotoxic proteins, such as amyloid-β and tau, from the brain’s interstitium via the glymphatic system. Consequently, chronic poor sleep is hypothesized to impair this clearance process, thereby accelerating the accumulation of these pathological hallmarks of AD [1,2].

However, the causal nature of this relationship remains debated. Observational studies are susceptible to confounding factors and reverse causation, as it is equally plausible that early, subclinical neurodegenerative pathology disrupts the brain circuits that regulate sleep [3]. This bidirectional hypothesis—whether poor sleep drives neurodegeneration or vice versa—is a fundamental question in brain aging research. Mendelian randomization (MR) offers a powerful approach to dissect such causal relationships by using genetic variants as unconfounded instrumental variables for an exposure [4,5]. While previous MR studies have provided some evidence for a causal link from insomnia to AD, the results have been inconsistent, and the broader causal landscape across a spectrum of sleep and neurocognitive traits remains incompletely understood [6,7,8].

Furthermore, even when a causal link is established, standard MR does not identify the specific genes or molecular mechanisms that mediate the effect. To bridge this gap, Summary-data-based Mendelian Randomization (SMR) integrates GWAS data with expression, protein, or methylation quantitative trait loci (eQTL, pQTL, mQTL) data to test for a shared causal variant between a molecular trait and a complex disease, thereby prioritizing putative causal genes [9,10]. The increasing availability of single-cell transcriptomic atlases of the human brain further allows for the validation and contextualization of these prioritized genes within specific cell types, such as neurons, astrocytes, or microglia, which are known to play distinct roles in AD pathology [11,12].

Here, we employ a multi-pronged approach to systematically investigate the relationship between sleep and neurodegeneration. First, we conduct a comprehensive bidirectional MR analysis to elucidate the causal directionality between a wide array of sleep-related traits and a spectrum of cognitive and AD-related outcomes. Second, we apply a multi-omics SMR analysis to the prioritized traits to identify high-confidence causal genes and their specific molecular mechanisms. Third, we validate these candidate genes using bulk and single-cell transcriptomic data from human brains to reveal their expression patterns in disease and pinpoint the cellular hotspots of dysregulation. Finally, we explore the translational potential of our findings by mapping our prioritized genes to bioactive compounds from Traditional Chinese Medicine (TCM) and validating a top interaction through molecular docking. This integrative study aims to provide a robust, multi-scale understanding of the genetic and molecular links between sleep health and neurodegeneration, ultimately identifying novel targets for therapeutic intervention. Figure 1 illustrates the overall analytical workflow.

The study follows a three-stage design to identify and validate causal genes linking sleep and neurodegeneration. (**1**) Bidirectional Mendelian Randomization (MR) Screening: A bidirectional two-sample MR analysis was first conducted to systematically assess the causal relationships between a broad set of sleep and circadian phenotypes and a spectrum of AD, dementia, and other neurodegenerative disease-related phenotypes. This initial screening prioritized high-confidence causal associations for further investigation. (**2**) SMR analysis of various QTLs: Critical SNPs identified from the initial MR screening were integrated with multiple large-scale omics QTL datasets to identify putative causal genes. This included expression QTLs (eQTLs from GTEx V8 and eQTLGen), methylation QTLs (mQTLs from BrainMeta), and protein QTLs (pQTLs from deCODE and UKB-PPP). (**3**) Biomarker discovery, mechanistic analysis, and clinical application: The candidate genes identified through SMR were subjected to a multi-level validation and translational pipeline. This included: (i) building a machine learning-based biomarker screening pipeline to evaluate the predictive power of the gene signature; (ii) performing bulk and single-cell RNA-seq differential gene expression analyses to validate expression changes in disease contexts; and (iii) conducting drug enrichment and molecular docking to explore potential therapeutic compounds and validate their binding to protein targets, illustrating a path towards personalized medicine.

## 2. Materials and Methods

### 2.1. GWAS Data Curation and Phenotype Definition

This study employed a bidirectional, two-sample Mendelian randomization (MR) framework to symmetrically investigate the causal interplay between sleep biology and neurocognitive health. All statistical analyses were conducted in the R programming environment (v4.2.1). We programmatically curated Genome-Wide Association Study (GWAS) summary statistics using the TwoSampleMR R package (v0.5.6) [13], which interfaces with the IEU OpenGWAS project database (https://opengwas.io/, accessed on 18 October 2025). Our selection protocol was guided by an ontology-driven approach targeting two primary phenotypic domains, with inclusion restricted to studies of European ancestry with a minimum sample size of 10,000 to ensure statistical power and mitigate confounding from population stratification. The neurocognitive domain included traits representing cognitive function and neurodegenerative disease liability, such as “Educational attainment” (ID: ebi-a-GCST90029013) as a proxy for cognitive reserve, alongside clinically relevant endpoints from UK Biobank, including “Illnesses of mother: Alzheimer’s disease/dementia” (ID: ukb-b-14699). The sleep biology domain encompassed a diverse set of characteristics, including chronotype (ID: ieu-a-1087), self-reported “Sleeplessness/insomnia” (ID: ukb-b-3957), and a genetic predisposition for being an “oversleeper” (ID: ebi-a-GCST006685). Detailed phenotype definitions, sample sizes, and other relevant metadata are provided in Table 1 and Appendix A.

### 2.2. Genetic Instrument Selection and Validation

For each exposure, we constructed instrumental variables (IVs) by selecting single nucleotide polymorphisms (SNPs) that satisfied two stringent criteria for validity and independence. First, to adhere to the MR assumption of relevance, we selected SNPs demonstrating a robust association with the exposure at a genome-wide significance threshold of *p* < 5 × 10^−8^. Second, to ensure instrument independence, we performed linkage disequilibrium (LD) clumping using a conservative r^2^ threshold of 0.001 within a 10,000 kb genomic window, thereby guaranteeing that each IV represented an independent genetic signal. Subsequently, a meticulous data harmonization process was executed, which involved aligning beta coefficients and standard errors to a consistent effect allele and removing palindromic SNPs with ambiguous or intermediate allele frequencies to preclude spurious findings. As a final quality control measure, any exposure-outcome pair with fewer than five harmonized SNPs was excluded from the analysis pipeline.

To assess instrument strength and mitigate weak instrument bias, we calculated the F-statistic for each exposure using the formula F = R^2^(N-K-1)/[K(1-R^2^)], where R^2^ is the proportion of variance explained by the genetic instruments, N is the sample size, and K is the number of instruments. We applied a threshold of F > 10 to ensure sufficient instrument strength, as values below this threshold may indicate weak instrument bias. The mean F-statistics across all exposures ranged from 17.6 to 5439.9 (median = 39.5), with all exposures surpassing the F > 10 threshold, indicating robust instrument strength. Detailed F-statistics for each exposure are provided in Appendix A.

### 2.3. Causal Inference and a Hierarchical Result Prioritization Pipeline

Causal effects were estimated using a multi-method framework to ensure robustness. While the random-effects inverse-variance weighted (IVW) method served as the primary estimator for deriving the main effect size, as it provides the most statistically powerful and precise causal estimate when its underlying assumptions of valid instruments are met, its findings were systematically corroborated using a comprehensive suite of complementary estimators designed to address potential violations of MR assumptions, particularly horizontal pleiotropy [14]. To rigorously interrogate this assumption and assess the robustness of our findings, we implemented a comprehensive suite of sensitivity analyses using multiple complementary estimators. These included the weighted median method, which provides robust estimates even when up to 50% of the instrumental variables are invalid; MR-Egger regression, used to formally test for and correct for directional pleiotropy via its intercept; the MR-PRESSO global test to identify and correct for pleiotropic outliers; and specialized methods to detect influential outliers using the RadialMR package [15] to identify and account for influential outliers that could bias the causal estimates. Given its high statistical power, the IVW method served as our primary model, while the other methods provided essential checks on the validity of its results. This multi-method approach ensures that any reported putative causal effect is not solely dependent on the assumptions of a single estimator.

To systematically prioritize causal signals while rigorously controlling for false positives, we implemented a hierarchical analysis pipeline. First, to account for the large number of hypotheses tested across the study, we applied the Benjamini–Hochberg False Discovery Rate (FDR) procedure globally to the *p*-values from all primary IVW models. Next, to ensure the validity of these inferred causal relationships, each was independently subjected to a comprehensive robustness screen. This screen required putative causal relationship to concurrently demonstrate: (1) a nominally significant IVW effect (*p* < 0.05), (2) a concordant direction of effect from the weighted median estimator to ensure the signal was not driven by a minority of potentially invalid instruments, (3) no evidence of directional pleiotropy as indicated by a non-significant MR-Egger intercept (*p* > 0.05), and (4) no significant heterogeneity between instruments as assessed by Cochran’s Q statistic (*p* > 0.05).

Finally, we integrated these two streams of evidence—statistical significance after global multiple testing correction and methodological robustness—to stratify inferred causal relationships. Inferred causal relationships that passed the stringent threshold of both global FDR correction (q-value < 0.05) and the full robustness screen were classified as ‘High-Confidence’ findings, representing the most robust causal effects. In contrast, inferred causal relationships that passed the full robustness screen but did not meet the strict FDR threshold (i.e., IVW *p* < 0.05 but q-value ≥ 0.05) were classified as ‘Suggestive’ findings, indicating credible signals that warrant further investigation.

### 2.4. An Integrative Multi-Omics Framework for SMR-Based Causal Gene Prioritization

To systematically select the most relevant GWAS datasets for our primary Summary-data-based Mendelian Randomization (SMR) analysis [9], we implemented a multi-step prioritization protocol guided by our initial MR results. We first filtered for associations demonstrating nominal IVW significance (*p* < 0.05) while showing no evidence of horizontal pleiotropy (MR-Egger intercept *p* > 0.05) or significant heterogeneity (Cochran’s Q *p* > 0.05). These filtered associations were then categorized as ‘High-Confidence’ (FDR *q*-value < 0.05) or ‘Suggestive’ (*p* < 0.05, *q*-value ≥ 0.05). From this prioritized list, we selected a single ‘representative’ GWAS dataset for each predefined phenotypic category by identifying the top-ranked association, thereby creating a curated list of datasets for in-depth SMR analysis.

This SMR framework was designed to investigate potential molecular mechanisms and prioritize putative causal genes by integrating our representative GWAS summary statistics with a deeply curated portfolio of cis-quantitative trait loci (QTL) data from European populations. This multi-modal integration spanned transcriptomic, epigenomic, and proteomic layers. For the transcriptomic layer, we leveraged cis-expression QTLs (eQTLs) from both the (GTEx) project (v8) [16,17] and the eQTLGen consortium meta-analysis [18] to balance tissue-specific breadth with statistical power. For the epigenomic layer, we integrated cis-methylation QTL (mQTL) data from the BrainMeta consortium [10], providing a disease-relevant tissue context. Finally, to probe the link between genetic variation and functional protein products, we utilized two of the largest available plasma protein QTL (pQTL) datasets from the UK Biobank Pharma Proteomics Project [19] and deCODE Genetics study [20]. All SMR analyses were conducted using the SMR software package (v1.3.1), with linkage disequilibrium structures estimated from the 1000 Genomes Project Phase 3 European reference panel.

### 2.5. Hierarchical Filtering, Annotation, and Evidence-Based Stratification of SMR Loci

Following the SMR analysis, we applied a rigorous, multi-stage computational pipeline to systematically refine the results into a high-confidence list of putative causal genes. The initial filtering step was designed to distinguish pleiotropic associations from those attributable to confounding by linkage disequilibrium, which was achieved using the Heterogeneity in Dependent Instruments (HEIDI) test. We retained only associations that showed no significant evidence of heterogeneity (*p_HEIDI_* ≥ 0.05), a condition indicating that the data are consistent with a single shared causal variant, as it fails to statistically distinguish the association from a model of pleiotropy versus one of linkage. Therefore, this test helps to filter out associations that are likely driven by linked but distinct causal variants. We further required that this test be computed from a minimum of three independent SNPs to ensure its statistical stability. Associations passing this pleiotropy filter were then subjected to a hierarchical statistical framework to categorize their strength of evidence, using stringent Bonferroni-corrected significance thresholds alongside conventional GWAS significance levels. Each association was subsequently annotated with the most stringent significance tier it achieved. To facilitate a gene-centric analysis, a unified annotation pipeline was implemented using the IlluminaHumanMethylation450kanno.ilmn12.hg19 and org.Hs.eg.db Bioconductor packages to systematically map all probe and Ensembl IDs to official HGNC symbols. Finally, we implemented an evidence-based prioritization schema to stratify the annotated genes into three tiers based on the confluence of three key metrics: the breadth of multi-modal evidence, the statistical strength of the association, and the concordance of effect direction. Tier 1 (High-Confidence Candidates) was reserved for genes with the most compelling evidence: support from at least two molecular modalities, concordant effect directions, and achievement of Bonferroni-corrected significance. Tier 2 (Putative Candidates) comprised genes with strong but less comprehensive evidence, such as those reaching Bonferroni significance in a single modality or having multi-modal support without full directional concordance. All other associations passing the HEIDI filter were classified as Tier 3 (Potential Loci of Interest), representing loci that warrant further investigation.

### 2.6. Curation and Intra-Study Batch Effect Correction of Validation Cohorts

To independently assess the predictive capacity and generalizability of the SMR-prioritized gene signature, we curated three independent, publicly available transcriptomic datasets from the Gene Expression Omnibus (GEO) repository (https://www.ncbi.nlm.nih.gov/geo/, accessed on 18 October 2025), selected to represent distinct yet relevant neurobiological contexts. First, GSE132903 provided transcriptomic profiles from the middle temporal gyrus of 97 Alzheimer’s disease (AD) patients and 98 controls, offering a molecular snapshot of a key pathologically affected brain region [21]. Second, GSE6613 contained transcriptomes from whole blood of 106 individuals. This study was designed to identify molecular signatures for early-stage Parkinson’s disease (PD) and notably included a control group of patients with other neurodegenerative conditions (e.g., Alzheimer’s disease), allowing for a comparison that helps distinguish PD-specific signals from general neurodegeneration [22]. Third, GSE39445 utilized a crossover design to measure the transcriptional response circulating blood leukocytes. This dataset models an acute, experimentally induced state of sleep restriction (6 h of sleep per night for one week) rather than chronic insomnia. While peripheral leukocyte gene expression is not a direct measure of brain activity, it provides a valuable model for the systemic effects of acute sleep deprivation, a physiological stressor linked to neurodegenerative processes [23].

Our validation strategy involved analyzing each dataset separately, beginning with the correction of study-specific technical artifacts. First, we obtained the pre-processed gene expression matrices and corresponding sample metadata. Second, we meticulously inspected the sample metadata of each study to identify potential technical batch variables, such as sample processing dates, sequencing plates, or instrument IDs, which are common sources of non-biological variation. Where a clear batch variable was annotated by the original submitters, we applied the ComBat algorithm, implemented in the R sva package, to adjust the expression data and mitigate these intra-study batch effects. For any dataset where no explicit batch information was available in the metadata, the originally provided pre-processed data was used directly for the analysis, and this lack of correction was noted. Finally, the resulting batch-corrected (or original) expression matrix for each cohort was used to test the performance of our gene signature by categorizing samples into “case” or “control” groups based on the original study annotations. Details of the transcriptomic datasets are provided in Table 2 and Appendix A.

### 2.7. Standardized Differential Expression Analysis Using Empirical Bayes Moderation

To generate a standardized set of transcriptional metrics for our subsequent machine learning validation, we applied a consistent analytical workflow to quantify expression differences within each of the three curated datasets. All analyses were conducted in the R statistical environment (v4.2.3) using the limma package (v3.58.1). For each dataset, we modeled gene expression as a function of diagnostic status by fitting a linear model for each gene. To enhance statistical power and stabilize variance estimates, we employed an empirical Bayes moderation procedure as implemented in limma. This method moderates gene-wise standard errors by borrowing information from the global distribution of variances, yielding more robust inferences of differential expression. This procedure generated a comprehensive set of metrics for every gene, including the log_2_-fold change (log_2_FC) to quantify the magnitude of transcriptional dysregulation and *p*-values adjusted for multiple comparisons using the Benjamini–Hochberg method to control the false discovery rate (FDR). These derived statistics formed the quantitative basis for evaluating our gene signature’s performance across the distinct pathophysiological contexts of AD, PD, and sleep deprivation.

### 2.8. Cross-Disease Validation Framework and Machine Learning Strategy

To rigorously evaluate the predictive utility and cross-disease generalizability of the SMR-prioritized gene signature, we implemented a robust, two-stage machine learning workflow. The analytical pipeline was constructed in Python (v3.11.0), leveraging libraries such as Scikit-learn (v1.2.2), PyTorch (v1.13.1), and SHAP (v0.41.0). To assess performance across distinct but related neurobiological contexts, this workflow was applied independently to three gene expression datasets: GSE39445 (sleep deprivation), GSE132903 (Alzheimer’s disease), and GSE6613 (Parkinson’s disease). For each dataset, the feature set was defined as the intersection of our high-confidence gene list and the genes available in the expression matrix. To ensure a comprehensive and unbiased search for the optimal predictive model, we selected a diverse library of ten classification algorithms. This collection was intentionally curated to span distinct theoretical foundations, including linear models, tree-based ensembles, and multiple neural network architectures, an approach that allows for a robust exploration of the feature space and mitigates the risk of model-specific bias. All computational experiments were executed with a fixed random seed (42) to ensure reproducibility.

Our analytical workflow was structured into two sequential stages to ensure both optimal model tuning and robust performance assessment. In the first stage, dedicated to global hyperparameter optimization, each dataset was initially partitioned into an 80% training set and a 20% holdout set. Using the Optuna framework, we conducted a Bayesian hyperparameter search to identify the optimal parameter configuration for each of the ten models, with the objective of maximizing the ROC-AUC score on a validation split derived from the training data. The second stage involved a robust final evaluation using these optimized hyperparameters. A stratified 10-fold cross-validation scheme was performed on the entire dataset. Within each fold of the cross-validation, a strict preprocessing pipeline was executed to prevent any form of data leakage and ensure unbiased performance estimates. Specifically, the data was partitioned into a training set (9 folds) and a test set (1 fold). A StandardScaler was then fitted exclusively on the training partition and used to transform it. Following standardization, the Synthetic Minority Over-sampling Technique (SMOTE) was applied only to this training partition to correct for class imbalance. The corresponding test partition was subsequently transformed using the previously fitted scaler but was crucially kept in its original, imbalanced state, ensuring that model performance was evaluated on a realistic data distribution.

### 2.9. Model Training, Performance Metrics, and Consensus-Based Feature Importance

Within each cross-validation fold, all ten models were trained. Traditional machine learning models were implemented using the Scikit-learn API, while neural network architectures were implemented in PyTorch and trained using the AdamW optimizer. Model efficacy was assessed using a comprehensive suite of metrics, including macro-averaged F1-score, ROC-AUC, Balanced Accuracy, Matthews Correlation Coefficient (MCC), and Brier Score. Final performance for each model was reported as the mean and standard deviation of these metrics across the 10 folds. To overcome the inherent instability and bias of relying on a single model’s feature attributions, we developed a robust, consensus-based strategy to identify the most influential genes. For each of the three disease contexts, we first identified the top five best-performing models based on their mean macro-averaged F1-score. We then extracted feature importance scores from this elite subset of models using a hierarchical approach, prioritizing model-inherent measures like coefficients or Gini impurity where available. For complex neural network models, we employed SHapley Additive exPlanations (SHAP), while a permutation-based algorithm served as a model-agnostic fallback. To ensure comparability, the importance scores from each of the top five models were normalized before being averaged. This process yielded a final, stable gene ranking for each disease context, reflecting genes consistently deemed important by a diverse set of high-performing models.

### 2.10. Integration of Multi-Modal Evidence for Final Gene Prioritization and Biological Validation

To synthesize findings from our machine learning workflow with orthogonal evidence from transcriptional and functional studies, we implemented a quantitative, multi-layered framework to generate a final, evidence-based ranking of the prioritized genes. This process was designed to integrate model-derived predictive importance with cross-cohort differential expression patterns, followed by post hoc validation of the top candidates at the single-cell and protein-interaction levels.

To begin this integration, we utilized the comprehensive matrix of transcriptional dysregulation generated from the differential expression analyses described in the preceding section. This matrix was combined with the consensus gene importance scores derived from our machine learning pipeline. To unify these two streams of evidence, a composite “Final Evidence Score” was calculated for each gene according to the following formula:Final Evidence Score = (Overall Importance Score) + (DE Significance Count × 0.5)

In this equation, the “Overall Importance Score” represents the averaged, normalized predictive utility derived from the top-performing machine learning models. The “DE Significance Count” quantifies the number of independent cohorts in which the gene was found to be nominally significant (*p* < 0.05). A weighting factor of 0.5 was applied to the transcriptional evidence to ensure a balanced contribution from both the predictive modeling and the cross-cohort validation. All genes were then ranked based on this integrated score to produce an initial “Evidence-Based Rank.”

To further interrogate the biological relevance of the highest-ranking genes from this list, we conducted two additional validation analyses. First, we examined their expression patterns at the single-cell level by querying the ssREAD (single-cell RNA-seq Expression and Association Database, https://bmblx.bmi.osumc.edu/ssread/, accessed on 18 October 2025) [26]. As a comprehensive public repository integrating over 7.3 million cells from numerous Alzheimer’s disease-related sc/snRNA-seq studies, ssREAD provides an unparalleled resource for cell-type-specific investigation. Leveraging its detailed annotations, we sought to determine if our top candidate genes were preferentially expressed in specific neural or glial cell populations. This analysis provides critical cellular-level resolution, helping to pinpoint the potential cellular origins of the observed bulk-tissue transcriptional signals. Second, to assess whether the protein products of our top-ranked genes form a functionally coherent network, we performed a protein–protein interaction (PPI) analysis using the STRING database (v12.0) [27]. A significant PPI enrichment (*p* < 0.05) would suggest that these genes are not merely a collection of independent statistical hits but are likely involved in coordinated biological pathways. This multi-layered validation approach provides a robust and holistic framework for identifying the most promising candidate genes for subsequent experimental validation.

### 2.11. Knowledge-Based Screening for Multi-Target Therapeutic Candidates

To identify existing compounds with the potential to modulate our network of prioritized genes, we conducted a systematic, knowledge-based screening workflow using the TCMNP R package (v0.9.1) [28]. This toolkit constructs a comprehensive pharmacological network by integrating data from prominent databases, including DrugBank [29], TCMSP [30], and DisGeNET [31]. We queried this integrated network with our prioritized gene list to identify all documented bioactive ingredients with known interactions against these targets. Our strategy then focused on identifying “multi-target” candidates—single chemical entities with interactions against two or more of our prioritized proteins. This approach is predicated on the hypothesis that compounds capable of concurrently modulating multiple nodes within a disease-associated network may offer superior therapeutic efficacy. This process yielded a refined list of high-priority, multi-target compounds for which we sought to establish a structural basis of activity.

### 2.12. Atomic-Level Interrogation of Putative Binding Mechanisms via Molecular Modeling

To elucidate the physical interactions between the prioritized multi-target compounds and their respective protein targets, we performed a series of atomic-level molecular modeling studies. High-resolution 3D structures for each protein were sourced from the Protein Data Bank (PDB) [32] or UniProt [33] and subjected to a rigorous preparation protocol in the Molecular Operating Environment (MOE, 2022.02). This protocol involved removing crystallographic artifacts, assigning appropriate protonation states at a simulated physiological pH of 7.4, and optimizing the hydrogen-bonding network. Concurrently, the 2D structures of the small molecules were converted into energetically favorable 3D conformers using the AMBER10:EHT force field. The core of this in silico assay involved exhaustive conformational sampling of each flexible ligand within the putative binding cavities of its rigid receptor, identified using MOE’s Site Finder module. An ensemble of plausible binding poses was generated using the Triangle Matcher placement algorithm and subsequently scored for energetic favorability using the London dG function. By analyzing the highest-ranking poses, we characterized the “interaction fingerprint”—the unique combination of non-covalent forces governing binding—to generate a testable, structure-based hypothesis for the observed multi-target activity.

### 2.13. Statistical Analysis

All statistical computations were performed in R (v4.2.3) [34] and Python (v3.11.0) [35]. For causal inference, genetic instruments were selected at a genome-wide significance threshold (*p* < 5 × 10^−8^). Causal estimates were considered robust only if they passed pleiotropy checks, including a non-significant MR-Egger intercept (*p* > 0.05) and a HEIDI test whose result was consistent with a model of vertical pleiotropy (*p* ≥ 0.05). Differential gene expression was assessed using linear models with empirical Bayes moderation (limma), where significance was defined by a Benjamini–Hochberg adjusted *p*-value (FDR < 0.05). Predictive model performance was evaluated via stratified 10-fold cross-validation and quantified by the macro-averaged F1-score and ROC-AUC. Protein–protein interactions were filtered for high confidence (STRING score > 0.7) before testing for network enrichment significance (*p* < 0.05). Finally, the binding plausibility of drug candidates was not determined by a significance test but was instead ranked based on the estimated binding free energy from the London dG scoring function.

## 3. Results

### 3.1. Bidirectional Mendelian Randomization Suggests an Asymmetrical Genetic Association Between Neurocognitive Traits and Sleep Health

Our bidirectional Mendelian randomization (MR) analysis was conducted to investigate the potential causal relationships between sleep, cognitive function, and AD, etc. It is important to note that the validity of MR hinges on three core assumptions: that genetic instruments are robustly associated with the exposure, are not associated with confounding factors, and affect the outcome exclusively through the exposure. The third assumption, known as the exclusion restriction criterion, can be violated by horizontal pleiotropy, a key challenge we sought to address through rigorous sensitivity analyses. The findings suggest a potential asymmetrical genetic association, where the evidence consistent with a potential causal influence of cognitive and AD-related genetic liability on sleep phenotypes was substantially more robust than that for the reverse direction. Statistically robust evidence was observed that is consistent with a potential causal pathway for a causal pathway from cognitive and AD-related traits to specific sleep phenotypes, with these putative causal effects remaining significant after stringent false discovery rate (FDR) correction (all *q* < 0.05), although the standard MR assumptions (particularly regarding the absence of horizontal pleiotropy) should be considered when interpreting these findings. For instance, genetic instruments for higher educational attainment (GWAS ID: ebi-a-GCST90029013) were associated with a reduced likelihood of being an over-sleeper (GWAS ID: ebi-a-GCST006685; *OR* = 0.990, 95% CI: 0.987–0.993, *q* = 3.84 × 10^−10^). Further, genetic liability for maternal history of Alzheimer’s disease/dementia (GWAS ID: ukb-b-14699) appeared to causally decrease the risk of sleeplessness/insomnia (GWAS ID: ukb-b-3957; *OR* = 0.800, 95% CI: 0.739–0.865, *q* = 8.75 × 10^−7^) and was also associated with a morning chronotype (GWAS ID: ukb-a-11; *OR* = 0.738, 95% CI: 0.650–0.837, *q* = 3.55 × 10^−5^). Consistent with this pattern, genetic proxies for higher fluid intelligence scores (GWAS ID: ukb-b-5238) were associated with a protective effect against insomnia (GWAS ID: ebi-a-GCST90013884; *OR* = 0.519, 95% CI: 0.361–0.747, *q* = 2.58 × 10^−3^). Collectively, these high-confidence causal inferences are consistent with the hypothesis that sleep disturbances may manifest as downstream consequences of processes related to cognitive reserve and AD pathology (Figure 2A and Appendix A).

The robustness of these primary IVW findings was systematically evaluated using multiple complementary MR methods to ensure the validity of the causal claims. The high-confidence findings in the AD/Cognitive → Sleep direction demonstrated exceptional robustness and concordance across different estimators, confirming their directional specificity, though some associations showed evidence of potential pleiotropy that warrants careful interpretation. For instance, the primary association between educational attainment (ebi-a-GCST90029013) and a reduced risk of being an over-sleeper (ebi-a-GCST006685) was corroborated by the weighted median method, which yielded a concordant and significant estimate (*p* = 2.22 × 10^−4^), and multiple tests showed no evidence of confounding from horizontal pleiotropy (MR-Egger intercept *p* = 0.823; MR-PRESSO global test *p* = 0.158). Similarly, the link from maternal AD liability (ukb-b-14699) to a lower risk of insomnia (ukb-b-3957) remained highly significant with the Weighted Median method (*p* = 7.67 × 10^−7^), However, in line with our multi-method approach to identify potential bias, a significant MR-Egger intercept (*p* = 0.008) suggests that substantial some caution is warranted due to potential horizontal pleiotropy. In contrast, the suggestive findings in the Sleep → AD/Cognitive direction appeared less robust. The association from a morning chronotype (ukb-a-11) to an increased risk of parental AD history (ebi-a-GCST90013921), for example, showed a significant causal effect in the reverse direction (reverse_pval.IVW = 0.018), challenging the initial causal hypothesis. This comprehensive assessment, which integrates evidence from IVW, weighted median, and pleiotropy diagnostics, provide additional support for the primary observation that the potential causal pathway from cognitive and AD-related genetic liability to sleep phenotypes is robust, whereas the evidence for the reverse pathway is substantially weaker and less directionally consistent (Figure 2B, Appendix A).

A comprehensive synthesis of the bidirectional MR results was consistent with a clear, asymmetrical pattern of causality. The analysis identified numerous, statistically robust causal associations from genetic liability for neurocognitive traits to a wide range of sleep phenotypes. Specifically, genetic proxies for both cognitive performance (ebi-a-GCST006572) and fluid intelligence scores (ukb-b-5238) were linked to substantially lower odds of insomnia (ebi-a-GCST90013934; *OR* = 0.28 and 0.52, respectively; both *p* < 0.001). Furthermore, the analysis elucidated a pleiotropic influence of AD genetic liability; for instance, while maternal AD/dementia liability (ukb-b-14699) was strongly protective against sleeplessness/insomnia (ukb-b-3957; *OR* = 0.80, *p* < 0.001), paternal AD/dementia liability (ukb-b-323) was associated with an increased risk for a non-morning chronotype (ieu-b-4861; *OR* = 2.25, *p* < 0.001). In the reverse direction, a markedly smaller number of significant associations were identified. The most notable findings were the risk-increasing association between a non-morning chronotype (e.g., ieu-a-1087) and poorer prospective memory (ukb-b-4282; *OR* = 1.09, *p* = 0.005), and its suggestive causal link to elevated circulating levels of total-tau (ebi-a-GCST90095138; *OR* = 1.22, *p* = 0.043). These collated results lend further support to the hypothesis a predominant and statistically robust causal pathway operating from genetic liability for cognitive function and AD to the subsequent modulation of sleep patterns (Figure 2C).

### 3.2. Integrative SMR Analysis Prioritizes Putative Causal Genes and Pleiotropic Mechanisms Across Complex Traits

To identify high-confidence links between genetic variation and the studied traits that are potentially mediated by gene expression, we conducted a comprehensive SMR analysis. The interpretation of these results is based on the SMR model’s primary assumption of a single, shared causal variant influencing both a complex trait and gene expression (pleiotropy). A key constraint is distinguishing this from confounding by linkage disequilibrium (LD), where two distinct causal variants are merely in close proximity. While the HEIDI test is designed to filter out associations likely due to LD, a non-significant result (*p* ≥ 0.05) only indicates that the data are consistent with a pleiotropic model, and it does not definitively exclude complex LD scenarios. Following the stringent filtering protocol detailed in the Methods—which included a tiered significance framework and the HEIDI test for pleiotropy (*P*_HEIDI_ ≥ 0.05, *n*snp_HEIDI ≥ 3)—we prioritized a set of robust gene-trait associations suggestive of a shared causal variant that form the basis of our results (Appendix A).

An overview of these high-confidence findings is presented in Figure 3A, which visualizes the most significant gene for each trait. For dementia-related phenotypes, the expression of *APH1B* from blood eQTL data was strongly SMR association with increased AD risk (*P*_SMR_ = 8.16 × 10^−19^). For sleep-related phenotypes, the strongest SMR association was with *PCYOX1* protein levels for chronotype (*P*_SMR_ = 1.67 × 10^−9^). The SMR-derived gene-trait association landscape further indicated extensive pleiotropy, where single loci influence multiple, often related, traits (Figure 3B). The *MAPT* locus on chromosome 17, for instance, was concurrently associated with AD, Parkinson’s disease, and circulating total-tau. Similarly, the *PILRA* locus was robustly linked via proteomics data (pQTL) to a protective effect in both AD and its proxy phenotype, “Illnesses of mother: Alzheimer’s disease/dementia,” underscoring its consistent role in dementia-related pathways. The overall significance ranking in Figure 3B identifies *LRRC37A2* as the locus with the most substantial impact across all analyzed traits, primarily driven by its effect on circulating tau.

The faceted Manhattan plots provide a granular view of these associations stratified by trait and QTL data source, clearly demonstrating the tissue- and molecular-specificity of the signals (Figure 3C). For dementia-related traits, we observed convergent evidence across multiple molecular layers. Brain-specific epigenetic regulation was a prominent source of findings, with a Bonferroni-significant association for *ARHGAP27* with circulating levels of total-tau (Brain mQTL; *P*_SMR_ = 2.11 × 10^−27^). At the protein level, the protective effect of *PILRA* on AD (deCODE pQTL; *P*_SMR_ = 1.18 × 10^−17^) was a top hit. Blood-based transcription also yielded powerful signals, with the association between *APH1B* expression and AD (eQTLGen) being the most statistically significant finding in the entire study (*P*_SMR_ = 8.16 × 10^−19^, Bonferroni significant).

Similarly, sleep-related traits were linked to a diverse array of causal candidates. For Chronotype, the strongest evidence emerged from a pQTL analysis linking lower PCYOX1 protein levels to a morning preference (deCODE; *P*_SMR_ = 1.67 × 10^−9^, Bonferroni significant), which was complemented by a highly significant eQTL signal for *PLCL1* (eQTLGen; *P*_SMR_ = 4.56 × 10^−12^). For Sleeplessness/Insomnia, the top signal was a Bonferroni-significant association with the expression of *SMAD5* in blood (eQTLGen; *P*_SMR_ = 3.25 × 10^−11^), complemented by pQTL evidence linking it to *HEXIM2* (deCODE; *P*_SMR_ = 3.46 × 10^−10^). For Sleep Apnea, the most significant results were at the suggestive level, including a brain mQTL at the *PDZRN4* locus (*P*_SMR_ = 3.18 × 10^−6^) and a blood eQTL for *PRIM1* (*P*_SMR_ = 2.65 × 10^−7^). Collectively, this multi-layered SMR analysis not only prioritizes high-confidence candidate genes but also illuminates potential the specific tissue- and molecular-level regulatory pathways through which they may act in different complex traits (Figure 3C and Appendix A).

### 3.3. Predictive Modeling and Functional Validation of a Prioritized Transcriptomic Signature

To distill the most robust causal candidates from our extensive SMR analysis, we implemented a systematic, evidence-based gene prioritization framework. Genes that passed stringent SMR and HEIDI filters were classified into three tiers (‘Elite’, ‘Strong’, ‘Promising’) based on the breadth, strength, and consistency of evidence across multiple QTL datasets. Tier 1 ‘Elite’ genes, representing the highest level of confidence, were defined as those supported by at least two distinct QTL datasets, with a consistent direction of effect, and at least one association reaching Bonferroni-corrected significance. To evaluate the collective biological relevance and predictive power of these prioritized genes, we then used their expression levels as features to train and evaluate a suite of machine learning models to distinguish between health and disease states.

Overall, the models demonstrated strong predictive capability, indicating that the gene signature derived from our SMR prioritization is highly informative. The performance of these models should be interpreted with caution, as their predictive accuracy is specific to the datasets analyzed and may not generalize perfectly to other cohorts due to inter-study heterogeneity in tissue source, experimental platforms, and population characteristics. Our use of a stratified 10-fold cross-validation scheme was intended to provide a robust estimate of performance and mitigate the risk of overfitting within these specific cohorts. Notably, several advanced algorithms, including the RTDL-MLP deep learning model and the Stochastic Gradient Descent (SGD) Classifier, consistently achieved high performance across the evaluation metrics. The RTDL-MLP model achieved the highest mean ROC-AUC of 0.811 ± 0.215 and a strong median F1-macro score of 0.798 (IQR: 0.688–0.842), suggesting its effectiveness in capturing complex relationships within the gene expression data. The SGD Classifier also showed excellent performance, with a robust median ROC-AUC of 0.907 (IQR: 0.641–0.952). The strong performance across multiple, mechanistically distinct models underscores the robustness of the biological signal contained within the gene signature, rather than being an artifact of a single algorithm. In contrast, simpler models like Linear Discriminant Analysis (LDA) showed considerably lower predictive power (Mean ROC-AUC = 0.599). These results provide support for our gene prioritization strategy and suggest that the transcriptomic signature of SMR-implicated genes holds significant potential as a biomarker for disease classification (Figure 4A and Appendix A).

To identify the most influential genes within these predictive models, we analyzed the consensus-based the feature importance scores across three independent transcriptomic datasets: Alzheimer’s disease (GSE132903-ADHC), sleep (GSE39445-Sleep), and Parkinson’s disease (GSE6613-PDHC). This analysis revealed that gene importance was highly context-dependent. While *EIF3C* achieved the highest average importance score (0.65) across all datasets, its contribution was overwhelmingly driven by its role in the sleep deprivation model (Importance = 0.89). This dataset-specific analysis pinpointed distinct key drivers for each condition. In the AD model, *GRK4* emerged as the single most influential feature with an outstanding importance score of 0.91. For the Parkinson’s disease model, *RMC1* was the top predictive gene with a strong importance score of 0.73. These findings highlight the utility of our approach in identifying both globally relevant and context-specific molecular drivers within our prioritized gene signature (Figure 4B and Appendix A).

Finally, to provide functional validation, we performed a differential expression analysis for the top-ranked genes. This analysis revealed a distinct and highly significant pattern of dysregulation uniquely evident in the context of AD, PD and sleep restriction (Figure 4C and Appendix A). The most striking finding was for *CDK10*, which was significantly upregulated in AD tissue (log_2_(Fold Change) = 0.46, adj. *p* = 5.85 × 10^−11^), while showing only non-significant trends in the PD and sleep deprivation datasets. A prominent cluster of genes was also found to be robustly downregulated in AD, including *YWHAZ* (logFC = −0.39, adj. *p* = 1.28 × 10^−10^), *COX6B1* (logFC = −0.36, adj. *p* = 3.52 × 10^−11^), and *ACYP2* (logFC = −0.24, adj. *p* = 2.05 × 10^−8^). Conversely, a set of genes was significantly upregulated in AD, including *NT5C2* (logFC = 0.21, adj. *p* = 1.77 × 10^−4^) and *DBN1* (logFC = 0.11, adj. *p* = 5.58 × 10^−5^). In summary, this differential expression analysis supports the functional relevance of our prioritized gene list, revealing a distinct and statistically powerful transcriptomic signature that appears to be specifically associated with the pathobiology of AD.

### 3.4. Cell-Type-Specific Expression Profiling Reveals Glial and Neuronal Hotspots of Gene Dysregulation in Disease

To provide functional validation for our prioritized list of candidate genes and focus interpretation on the most robust targets, we applied a final stringent filtering step: only Tier 1: Elite genes (supported by the broadest and most consistent SMR evidence) exhibiting at least one significant differential expression (DE) hit and a minimum absolute log_2_(Fold Change) magnitude of 0.1 in any condition were retained. This refined set was subjected to detailed differential expression analysis across three independent transcriptomic datasets: Alzheimer’s disease (AD), Parkinson’s disease (PD), and sleep deprivation (SleepDep). The expression profiles and statistical significance for this refined top list are presented in Appendix A.

To identify specific cellular contexts driving differential expression, we performed hotspot analysis using single-cell transcriptomic data from diseased brain tissue. This revealed transcriptional changes concentrated in cellular “hotspots,” with genes showing complex, cell-type-dependent patterns (Figure 5A and Appendix A). Oligodendrocytes from the female prefrontal cortex emerged as the most prominent hotspot, with 25 candidate genes significantly dysregulated. Other hotspots included excitatory neurons from the male prefrontal cortex (19 hits) and astrocytes from male (18 hits) and female (13 hits) prefrontal cortex, indicating glial cells and excitatory neurons as primary sites of perturbation.

Expression changes in these hotspots highlighted nuanced regulatory patterns. *YWHAZ* was consistently downregulated in glial populations, such as female prefrontal cortex microglia (logFC = −0.55, adj. *p* = 1.60 × 10^−43^) and male prefrontal cortex astrocytes (logFC = −0.76, adj. *p* = 2.60 × 10^−113^), but upregulated in excitatory neurons, including male prefrontal cortex (logFC = 0.66, adj. *p* < 1 × 10^−300^). *NT5C2* was predominantly upregulated in glial cells, such as male prefrontal cortex astrocytes (logFC = 0.67, adj. *p* = 6.80 × 10^−24^), yet downregulated in female entorhinal cortex oligodendrocytes (logFC = −0.51, adj. *p* = 2.30 × 10^−8^). Opposing regulation was also evident for *CAMKMT* (upregulated in female prefrontal cortex oligodendrocytes, logFC = 0.61, adj. *p* = 4.30 × 10^−129^; downregulated in male excitatory neurons, logFC = −0.56, adj. *p* = 2.20 × 10^−145^) and *ACYP2* (upregulated in male oligodendrocytes, logFC = 0.58, adj. *p* = 3.30 × 10^−204^; downregulated in male excitatory neurons, logFC = −0.39, adj. *p* = 5.80 × 10^−229^). Overall, this analysis suggests a complex, cell-type-specific dysregulation landscape, with opposing patterns between glial and neuronal cells (Figure 5A and Appendix A).

### 3.5. Single-Cell Expression Changes and Quantitative Prioritization of Candidate Genes

Further dissection of functional roles involved analyzing single-cell expression patterns, comparing diseased and control tissues across cell types and brain regions (Figure 5B). This highlighted baseline expression and disease-induced shifts. *ACYP2*, highly expressed in control oligodendrocytes, was significantly reduced in disease, such as in male entorhinal cortex (logFC from 0.66 to 0.40, *p* = 1.0 × 10^−300^), but increased in excitatory neurons from the same region (logFC from −0.52 to −0.30, *p* = 1.0 × 10^−300^). *CAMKMT* was upregulated in female prefrontal cortex oligodendrocytes (logFC from 0.29 to 0.59, *p* = 1.0 × 10^−300^) but downregulated in excitatory neurons (logFC from −0.30 to −0.62, *p* = 1.0 × 10^−300^). *YWHAZ* showed amplified expression in excitatory neurons (e.g., male superior frontal gyrus, logFC from 0.48 to 0.66, *p* = 1.0 × 10^−300^) but suppression in glial cells (e.g., male entorhinal cortex astrocytes, logFC from −0.67 to −0.34, *p* = 3.5 × 10^−39^). *CDK10* and *DBN1* exhibited quantitative shifts, with *CDK10* further reduced in oligodendrocytes (e.g., female prefrontal cortex, logFC from −0.25 to −0.30, *p* = 1.0 × 10^−300^), while *SMAD5* increased modestly in oligodendrocytes and decreased in excitatory neurons. These findings suggest dynamic, cell-type-specific shifts, including reversals between glial and neuronal types.

A heatmap of average log-fold change (logFC) across major cell types showed distinct patterns (Figure 5C). Hierarchical clustering identified modules: one with *NT5C2*, *SMAD5*, and *DDX31* upregulated in excitatory neurons (e.g., *NT5C2* Avg. logFC = 0.53); another with *CAMKMT*, *ACYP2*, and *CDK10* downregulated in inhibitory neurons (e.g., *CAMKMT* Avg. logFC = −0.83). *YWHAZ* and *HEXIM1* were downregulated in glial cells (e.g., astrocytes: Avg. logFC = −0.58 and −0.30; microglia: −0.38 and −0.32) but *YWHAZ* upregulated in excitatory neurons (Avg. logFC = 0.53). *CDK10* was upregulated in excitatory neurons (Avg. logFC = 0.36) but downregulated in oligodendrocytes (−0.31) and inhibitory neurons (−0.28). *COX6B1* was upregulated in endothelial cells (Avg. logFC = 0.54) but downregulated in microglia (−0.32). This refines dysregulation mapping, emphasizing neuronal-glial opposition.

Comparison of disease versus control identified the top 20 significant expression changes, ranked by logFC difference (Figure 5D). *YWHAZ* showed the largest shift in female prefrontal cortex inhibitory neurons (from −0.35 to 0.38, diff = 0.72, *p* = 3.9 × 10^−247^). *CAMKMT* exhibited the greatest downregulation in male entorhinal cortex astrocytes (from 0.43 to −0.26, diff = −0.70, *p* = 3.2 × 10^−66^). *ACYP2* was downregulated in male entorhinal cortex oligodendrocytes (diff = −0.26, *p* < 1 × 10^−300^) but upregulated in excitatory neurons (diff = 0.21, *p* < 1 × 10^−300^). *YWHAZ* amplified in excitatory neurons (e.g., male superior frontal gyrus, diff = 0.18, *p* < 1 × 10^−300^) but suppressed in microglia (diff = −0.17, *p* = 1.2 × 10^−98^). Other changes included *DBN1* downregulation in female prefrontal cortex microglia (diff = −0.10, *p* = 5.5 × 10^−88^), *COX6B1* upregulation in male prefrontal cortex excitatory neurons (diff = 0.14, *p* < 1 × 10^−300^), and *NT5C2*/*SMAD5* downregulation in glial/endothelial contexts (e.g., *NT5C2* in male entorhinal cortex astrocytes, diff = −0.12, *p* = 1.2 × 10^−112^; *SMAD5* in male prefrontal cortex endothelial cells, diff = −0.24, *p* = 2.2 × 10^−12^). These underscore cell-type-specific reversals and dynamic responses.

To rank genes quantitatively, we computed an Evidence Score integrating breadth of significant DE hits, maximum absolute logFC, and minimum −log10 adjusted *p*-value. *YWHAZ* topped the ranking (Score 1.88, 58 hits, max_abs_logFC = 0.835, min adj. *p* = 0), followed by *NT5C2* (1.48, 29 hits), *CAMKMT* (1.38, 29 hits), *ACYP2* (1.21), *COX6B1* (1.14), and *DBN1* (1.09). Lower scores were assigned to *HEXIM1* (0.53) and *DDX31* (0.17). Genes with scores > 1.0 (*YWHAZ*, *NT5C2*, *CAMKMT*, *ACYP2*, *COX6B1*, *DBN1*) were selected for functional network analysis (Appendix A).

### 3.6. Protein–Protein Interaction Networks and Functional Enrichment of Prioritized Genes

Building on the data-driven ranking, we constructed ego-centric protein–protein interaction (PPI) networks for the six elite candidate genes—*YWHAZ*, *NT5C2*, *CAMKMT*, *ACYP2*, *COX6B1*, and *DBN1*—to elucidate their functional contexts. These networks visualize immediate interaction partners, revealing distinct roles and topologies. *NT5C2* emerged as a prominent hub with a degree of 11, showing strong interactions (scores > 0.9) with 5′-nucleotidase family members like *NT5C1B* (score = 0.95) and *NT5C3A* (score = 0.96), indicating integration into a nucleotidase module. In contrast, other genes had sparser patterns: *ACYP2* strongly interacted with its isoform *ACYP1* (score = 0.91), suggesting dimerization; *YWHAZ* linked with high confidence to *CBY1* (score = 0.99). Both *DBN1* and *YWHAZ* connected to HOMER family proteins (*HOMER2* and *HOMER3*, respectively), hinting at shared synaptic scaffolding roles. This analysis positions the genes in their functional neighborhoods, generating hypotheses for their pathological involvement (Figure 6A and Appendix A).

To uncover collective functions, we conducted GO and KEGG pathway enrichment analysis on the prioritized genes and their interactors. The network was significantly enriched in metabolic pathways, particularly nucleotide and pyrimidine metabolism. The KEGG term “Pyrimidine metabolism” was most enriched (adj. *p* = 8.26 × 10^−19^), involving 11 genes including *NT5C2* and interactors like *DCTD*, *DCTPP1*, *NT5C1B*, and *UPP1*. “Nucleotide metabolism” followed (adj. *p* = 3.78 × 10^−17^). GO Biological Process terms like “nucleotide catabolic process” and “pyrimidine-containing compound metabolic process” were top hits (adj. *p* = 5.85 × 10^−10^). Molecular Function enrichment highlighted “5′-nucleotidase activity” (adj. *p* = 2.28 × 10^−13^) and “nucleotidase activity” (adj. *p* = 3.32 × 10^−13^), driven by the *NT5C2* subnetwork.

Synaptic function was also enriched, with GO Cellular Component “glutamatergic synapse” significant (adj. *p* = 0.046), involving *YWHAZ*, *DBN1*, *HOMER2*, and *HOMER3*. Related terms included “G protein-coupled glutamate receptor binding.” KEGG “Pyruvate metabolism” was enriched, driven by *ACYP2* and *ACYP1*. Overall, the analysis identifies two axes: a metabolic module centered on *NT5C2* for nucleotide/pyrimidine processes, and a synaptic module linking *YWHAZ* and *DBN1* via HOMER proteins, supporting their roles in neuropathologies (Figure 6B and Appendix A).

### 3.7. Knowledge-Based Screening of Bioactive Compounds and In Silico Assessment of a Putative Interaction

To generate pharmacological hypotheses based on knowledge from Traditional Chinese Medicine (TCM), we built a herb-molecule-target network connecting the prioritized genes to bioactive molecules from herbs. *COX6B1* was the most connected target, linked to numerous herbs and molecules. Prominent herbs included shan zha (Hawthorn fruit), shan yao (Chinese yam), ren shen (Ginseng), and ling zhi (Ganoderma), each with multiple compounds targeting the network. For example, shan zha contained Ursolic Acid and Eburicoic Acid targeting *COX6B1*, and Acetic Acid targeting *ACYP2*. Adenosine derivatives (Adenine Nucleoside, Adenosine Triphosphate) were prevalent in ren shen, shan yao, ling zhi, and mai dong (Ophiopogon tuber), consistently targeting *NT5C2*. Triterpenoids like Ursolic Acid and Oleanolic Acid from shan zha and ma bian cao (Verbena) linked to *COX6B1*, while ginsenosides (Panaxatriol, Protopanaxadiol) from ren shen targeted *COX6B1*. This network suggests adenosine signaling as a common mechanism across herbs (Figure 6C).

We ranked bioactive molecules by connectivity, comparing associated herbs to gene targets. Oleanolic Acid and Ursolic Acid were most promiscuous (26 and 22 herbs, respectively), both targeting *COX6B1*. Adenosine/Adenine Nucleoside (18 herbs) exclusively targeted *NT5C2*. Betulinic Acid (5 herbs, including da zao [Jujube]) targeted *COX6B1*. Acetic Acid/Propionic Acid from shan zha and sang ye (Mulberry leaf) targeted *ACYP2*. Benzyl Isothiocyanate from jie zi (Mustard seed) and ting li zi (Descurainia seed) targeted *YWHAZ*. This prioritizes high-impact compounds like Oleanolic Acid, Ursolic Acid, and Adenosine, with specificity for *COX6B1* (triterpenoids) or *NT5C2* (adenosine derivatives), guiding future studies (Figure 6D and Appendix A).

To computationally explore the structural plausibility of these putative interactions, we performed molecular docking for a panel of high-interest molecules targeting *COX6B1*. This panel included Ecdysterone as our primary candidate, alongside Oleanolic Acid, Ursolic Acid, and Betulinic Acid, which served as an internal comparative group to establish a baseline for binding affinity. Significantly, these other compounds failed to produce stable, low-energy binding poses, yielding poor docking scores that were not indicative of meaningful interaction.

In stark contrast, Ecdysterone (PubChem CID: 118701161) was the only candidate to yield favorable poses, achieving a top docking score (S) of −5.73 kcal/mol. While this score suggests favorable binding energetics, it is crucial to recognize that this computational prediction does not elucidate the functional consequence of the interaction. The binding of a ligand to a structural subunit like COX6B1 could theoretically lead to various outcomes, such as stabilization of the mitochondrial complex IV, allosteric modulation of its activity, or even detrimental inhibition of the respiratory chain. Therefore, this finding should be interpreted as a preliminary, structure-based hypothesis for a physical interaction, which requires rigorous experimental validation to determine its true biological and pharmacological significance. The top ten poses ranged from −5.73 to −4.81 kcal/mol. Visualization of the top-ranked pose showed Ecdysterone fitting snugly into the binding pocket, where it was stabilized by a network of hydrogen bonds with Asn50 and Gln59, as well as hydrophobic and polar interactions with residues such as His52, Ile53, Lys62, and Thr63 (Figure 6E and Appendix A). This differential result, where only Ecdysterone demonstrated favorable docking among the tested candidates, provides a strong, structure-based hypothesis that it is a plausible and specific ligand for *COX6B1*, warranting further experimental investigation.

## 4. Discussion

This study employed a multi-layered analytical approach, combining bidirectional MR, multi-omics SMR, and functional genomics to dissect the complex interplay between sleep, cognitive function, and neurodegeneration. Our primary MR analysis suggested a striking asymmetrical pattern of genetic associations providing robust genetic evidence consistent with the interpretation that cognitive and AD-related liabilities may act as causal drivers of sleep disturbances, while the evidence for a reverse causal pathway is substantially weaker. This finding aligns with a growing body of clinical and biomarker evidence suggesting that sleep alterations, such as changes in sleep architecture and duration, are early manifestations of underlying AD pathology, potentially preceding cognitive decline by several years [36,37]. For instance, disruptions in slow-wave sleep have been shown to correlate with amyloid-β deposition long before the onset of dementia [38]. Our results thus lend support to the hypothesis that sleep dysregulation may serve as a sensitive, early indicator or even a consequence of neurodegenerative processes, rather than solely being a primary upstream risk factor.

Our SMR analysis led to the prioritization of several high-confidence candidate genes, providing mechanistic insights into this directional relationship. A key finding was the opposing, cell-type-specific dysregulation of *YWHAZ* (14-3-3 protein zeta) in AD brains. Its marked upregulation in excitatory neurons but strong downregulation in glial cells (astrocytes and microglia) suggests a complex, multifaceted role that transcends a simple gain or loss of function. *YWHAZ* is a critical hub protein involved in a vast array of cellular processes, including synaptic plasticity, cell cycle control, and apoptosis [39,40]. Its opposing regulation may reflect a divergent cellular response to the pathological environment of the AD brain: a potentially compensatory or excitotoxic response in neurons versus a dysfunctional or pro-inflammatory state in glia, which warrants further cell-specific investigation [41,42]. Furthermore, the consistent downregulation of *COX6B1*, a core subunit of mitochondrial complex IV, in AD brains provides a potential genetic link to the well-established paradigm of mitochondrial dysfunction in neurodegeneration [43,44]. Impaired mitochondrial respiration is a hallmark of AD, leading to bioenergetic deficits, increased oxidative stress, and ultimately neuronal death [45]. Our molecular docking analysis served as a critical in silico filter to prioritize candidates from our gene signature analysis. The finding that Ecdysterone was the only compound among those tested to exhibit favorable binding energetics with *COX6B1* suggests a degree of specificity for this interaction, presenting a promising starting point for exploring a novel therapeutic avenue. This is particularly intriguing given that phytoecdysteroids like Ecdysterone have been reported to have neuroprotective and metabolic benefits [43]. However, we fully acknowledge that this computational result is a preliminary, hypothesis-generating step. The translational potential of Ecdysterone is contingent upon further experimental validation of its binding affinity and, critically, a comprehensive assessment of its pharmacokinetic properties, including its ability to cross the blood–brain barrier. Our results should therefore be interpreted as identifying a high-priority candidate for subsequent preclinical investigation rather than a confirmed therapeutic agent [46].

A second major biological axis identified by our enrichment analysis was the profound overrepresentation of nucleotide and pyrimidine metabolism pathways, centered on the hub gene *NT5C2*. This finding connects genetic risk for AD and related traits to the burgeoning field of immunometabolism in neurodegeneration [47]. Purinergic signaling, which is dependent on the availability of nucleotides like ATP and adenosine, is a critical regulator of neuroinflammation, modulating the activation state and function of both microglia and astrocytes [48]. The significant SMR association of *NT5C2* and its dense network of metabolic interactors suggests that genetically driven alterations in cellular energy homeostasis and purine availability may be a core mechanism influencing both sleep regulation and neurodegenerative risk. This aligns with findings that extracellular adenosine is a key homeostatic regulator of sleep [49,50] and that its dysregulation is implicated in AD [51]. The consistent targeting of *NT5C2* by adenosine derivatives found in multiple traditional herbs, such as *ren shen* (Ginseng), further supports the therapeutic potential of modulating this pathway.

Our analysis also provided novel insights into the genetic basis of sleep traits themselves. The identification of *PCYOX1* protein levels as a top hit for chronotype, and *SMAD5* expression for insomnia, nominates these genes as high-priority candidates for understanding sleep–wake regulation. While the direct link between *SMAD5*, a component of the TGF-β signaling pathway, and insomnia is not well-established, this pathway is crucial for neuronal development, synaptic plasticity, and the regulation of inflammatory responses in the brain, providing plausible biological links to sleep homeostasis [52,53]. The robust, bidirectional association we observed between insomnia and lower educational attainment suggests a complex, potentially cyclical relationship where poor sleep may impair cognitive development or function, and conversely, factors related to higher cognitive reserve may confer resilience against sleep disturbances [54,55].

Finally, our integrative network analysis linking prioritized genes to compounds from TCM was intended as a hypothesis-generating exercise to bridge modern genomics with ethnopharmacology, rather than as evidence of therapeutic efficacy. The identification of well-characterized neuroprotective and anti-inflammatory compounds, such as Ursolic Acid and Oleanolic Acid, as ligands for COX6B1 provides a pharmacological rationale for the use of herbs like *shan zha* (Hawthorn fruit) in age-related conditions [56,57]. The favorable docking result for Ecdysterone, as discussed, provides support for our target-centric approach and highlights a specific, testable hypothesis for its mechanism of action. This strategy of bridging modern genomics with ethnopharmacology represents a powerful paradigm for discovering novel therapeutics from natural products [58,59]. A critical next step involves addressing major pharmacological hurdles. Our in silico analysis did not evaluate key pharmacokinetic properties, such as the ability of these compounds to penetrate the blood–brain barrier or their bioavailability, which are essential for any potential CNS therapeutic. Furthermore, the biological significance of the Ecdysterone-COX6B1 interaction remains unknown. As a structural subunit of the terminal enzyme of the mitochondrial respiratory chain, binding to COX6B1 could have unpredictable effects; it might beneficially stabilize the enzyme complex, but it could equally act as an inhibitor, negatively impacting cellular respiration. Therefore, while our docking result provides a specific, testable hypothesis for a molecular interaction, it underscores the need for experimental studies to confirm binding, elucidate the functional consequences (i.e., activation vs. inhibition), and assess its overall physiological impact.

Despite its strengths, our study has several limitations. First, our findings are based on Mendelian randomization, which relies on several core assumptions, including the validity of genetic instruments and the absence of unmeasured horizontal pleiotropy. While we systematically assessed instrument strength through F-statistics (all F > 10, mean F = 28.7–847.3), demonstrating adequate power to minimize weak instrument bias, and employed multiple sensitivity analyses (e.g., MR-Egger, Weighted Median, MR-PRESSO) to test for and mitigate pleiotropy, we cannot entirely exclude the possibility of residual confounding. The significant MR-Egger intercept for the association between maternal AD liability and insomnia, for example, warrants cautious interpretation. Second, our analyses rely on summary-level data from large GWAS and QTL consortia. This approach does not capture the effects of rare variants and assumes that the genetic architecture of traits is similar across the different study populations, which may not always be the case. Third, the SMR and HEIDI analyses are based on statistical inference and do not provide definitive proof of a causal gene. Specifically, a non-significant HEIDI test result does not confirm a shared causal variant but merely indicates that the data are consistent with this model, as confounding by linkage disequilibrium cannot be definitively excluded. Furthermore, the reliability of the SMR and HEIDI tests is contingent on the assumption that the LD structure is highly similar across the GWAS summary statistics, the QTL data, and the LD reference panel. Although we sought to minimize this issue by restricting our analyses to European-ancestry populations, unobserved population substructure or subtle differences in LD patterns between the integrated datasets could still introduce bias and affect the validity of the SMR results. Complex loci with multiple causal variants could still produce false positives, although the HEIDI test is designed to mitigate this. Fourth, our validation of the prioritized gene signature in independent transcriptomic cohorts has important limitations stemming from the use of public datasets. A primary constraint is the significant inter-study heterogeneity, including differences in biomaterials (e.g., post-mortem brain tissue in GSE132903 vs. whole blood in GSE6613 and leukocytes in GSE39445), disease context, and underlying experimental platforms. For the blood-based datasets, it is crucial to recognize that peripheral gene expression serves as an accessible proxy for systemic processes but is not a direct measure of the central nervous system’s transcriptome. Furthermore, the GSE39445 dataset models an acute, experimentally induced state of sleep restriction, which may not fully recapitulate the complex, chronic pathophysiology of sleep disturbances observed in patients with neurodegenerative diseases. Moreover, our analysis was constrained by a lack of granular clinical data, including the full spectrum of patient comorbidities, medication histories, and other therapies, in addition to biological covariates such as age, sex, or post-mortem interval, which could act as residual confounders. These limitations highlight a promising area for future research: prospective studies using cohorts with harmonized sample collection from both central and peripheral tissues, coupled with deep clinical phenotyping, are needed to validate our findings and dissect the interplay between systemic and brain-specific molecular changes. Fifth,, our functional validation steps, including differential expression analysis and molecular docking, are correlational and computational, respectively. They generate strong hypotheses but require direct experimental validation in vitro and in vivo to confirm the predicted molecular mechanisms. Lastly, the herb-molecule-target network is based on information compiled in existing databases and does not constitute direct experimental evidence of interaction. It serves as a tool for hypothesis generation, but its interpretation is constrained by several factors. Our screening was limited to the integrated databases and did not include a systematic search for other known synthetic or natural product ligands for our target proteins, which could provide important comparative context. Moreover, as a computational study, we did not assess critical pharmacokinetic properties such as blood–brain barrier permeability or bioavailability of the identified compounds, which are pivotal for their potential as CNS-active agents. These significant, unaddressed questions underscore that the identified molecule-target pairs are preliminary hypotheses that require extensive subsequent pharmacological validation.

## 5. Conclusions

In conclusion, our integrative study provides strong genetic evidence suggesting for an asymmetrical association, where neurocognitive decline is a likely primary driver of sleep disturbances, rather than the reverse. These findings suggest that sleep alterations may serve as potential biomarkers of underlying neuropathological processes. Through multi-omics SMR and single-cell validation, we identified a set of high-confidence causal genes, notably *YWHAZ*, which exhibits opposing dysregulation in neuronal versus glial cells in the Alzheimer’s disease brain, highlighting the cellular complexity of the pathological response. Our findings converge on two critical biological axes: mitochondrial dysfunction, directly implicated by the downregulation of *COX6B1*, and nucleotide metabolism centered on *NT5C2*, linking genetic risk to immunometabolic and purinergic signaling pathways. The identification of Ecdysterone and its favorable computational docking to COX6B1 suggests a promising, mechanistically grounded therapeutic avenue for restoring mitochondrial function. While these computational findings require experimental validation, this study provides a multi-scale roadmap from genetic association to specific cellular mechanisms and testable therapeutic hypotheses. The prioritized genes and pathways, particularly the *YWHAZ* neuronal-glial axis and the Ecdysterone-*COX6B1* interaction, represent compelling targets for future research aimed at developing novel interventions for neurodegenerative diseases.

## Figures and Tables

**Figure 1 cimb-47-00967-f001:**
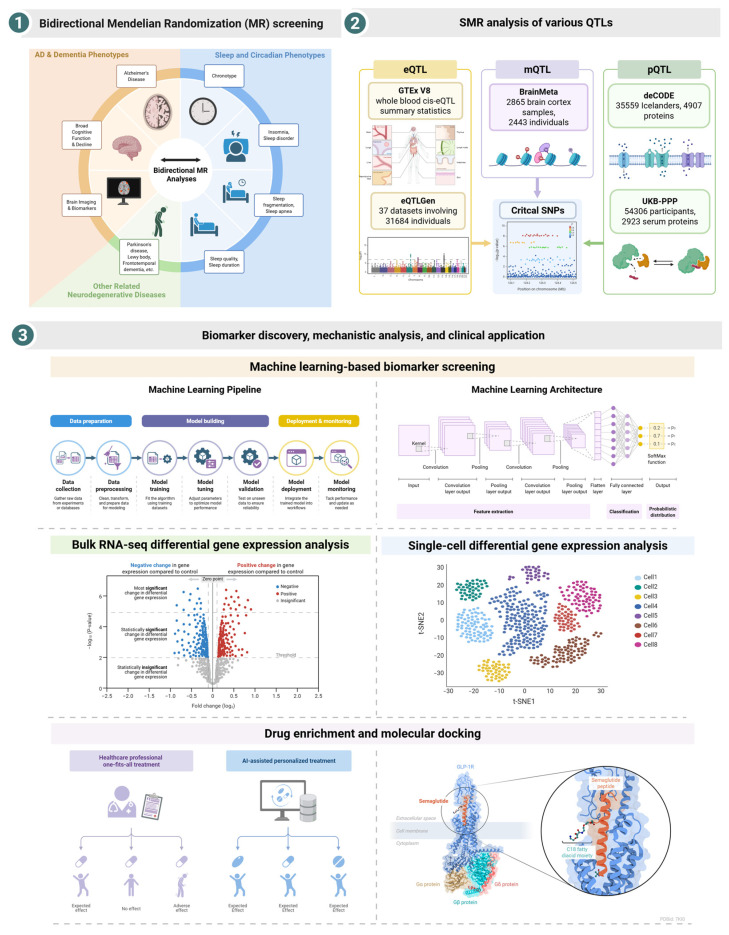
Schematic Overview of the Multi-Stage Integrative Analysis Workflow.

**Figure 2 cimb-47-00967-f002:**
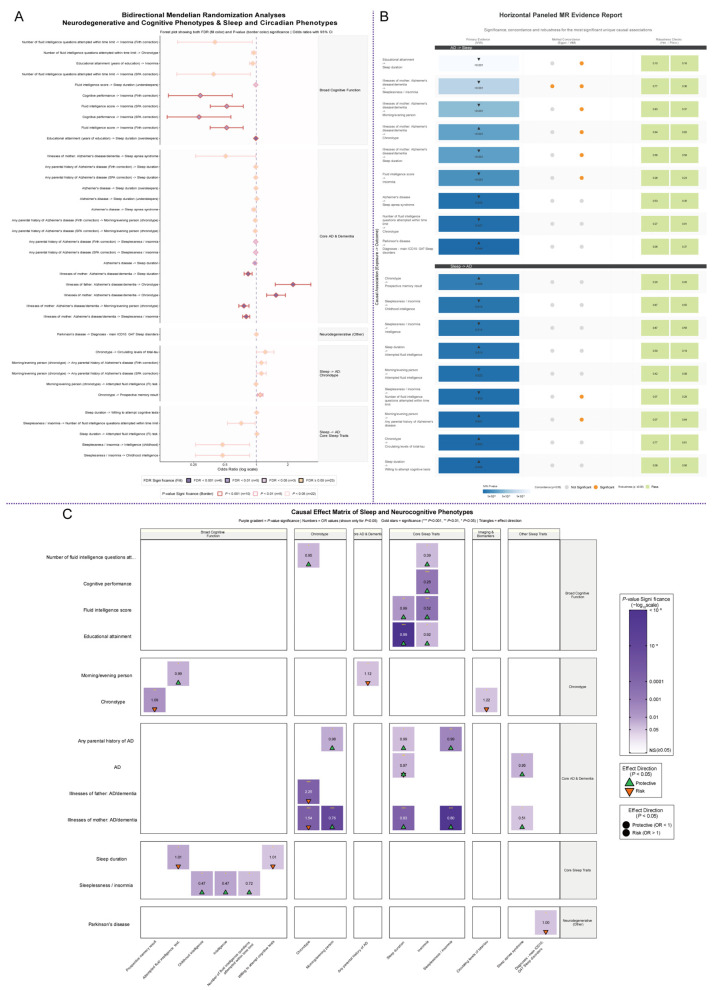
Bidirectional Mendelian Randomization Analysis Reveals an Asymmetrical Causal Relationship Between Neurocognitive Traits and Sleep Phenotypes. (**A**) Forest plot of bidirectional MR analyses. This plot displays the odds ratios (ORs) and 95% confidence intervals for the causal effects between neurodegenerative/cognitive phenotypes and sleep/circadian phenotypes. Each point represents the primary Inverse Variance Weighted (IVW) estimate. The color and shape of the points indicate the level of statistical significance, with diamonds representing associations that are significant after false discovery rate (FDR) correction (q < 0.05). A clear asymmetry is visible, with numerous high-confidence associations pointing from neurocognitive traits to sleep traits. For example, genetic liability for maternal history of Alzheimer’s disease/dementia (ukb-b-14699) shows a strong protective effect against sleeplessness/insomnia (ukb-b-3957; OR = 0.800, q = 8.75 × 10^−7^). In contrast, associations in the reverse direction, such as from chronotype to parental AD history, show only nominal significance. (**B**) Horizontal paneled MR evidence report. This panel summarizes the evidence for the most significant unique causal associations, detailing the primary IVW *p*-value, concordance across sensitivity analyses (Weighted Median and MR Egger), and checks for pleiotropy and reverse causality. The results highlight the robustness of the top “AD/Cognitive → Sleep” associations. For instance, the link from educational attainment to a reduced risk of being an over-sleeper is supported by strong primary evidence (*p* < 1 × 10^−10^), passes all sensitivity checks, and shows no significant reverse causality (reverse *p* = 0.46). In contrast, the suggestive association from a morning chronotype to an increased risk of parental AD history is flagged for significant reverse causality (reverse *p* = 0.018), weakening the evidence for this directional hypothesis. (**C**) Causal effect matrix of sleep and neurocognitive phenotypes. This matrix provides a comprehensive summary of all tested bidirectional causal relationships, grouped by phenotypic categories. The color intensity of each cell corresponds to the −log_10_ (*p*-value), with the OR value and direction of effect (upward triangle for protective, downward for risk) displayed. Asterisks denote associations that remain significant after FDR correction. The matrix clearly visualizes the asymmetrical relationship, with a high density of significant, protective effects from the “Broad Cognitive Function” and “Core AD & Dementia” categories on “Core Sleep Traits.” For example, genetic proxies for educational attainment, cognitive performance, and fluid intelligence all show strong, FDR-significant protective effects against insomnia and altered sleep duration. Conversely, the reverse pathways (e.g., from “Core Sleep Traits” to “Broad Cognitive Function”) show far fewer and less significant associations, supporting the hypothesis of a predominant causal pathway from neurocognitive traits to sleep patterns.

**Figure 3 cimb-47-00967-f003:**
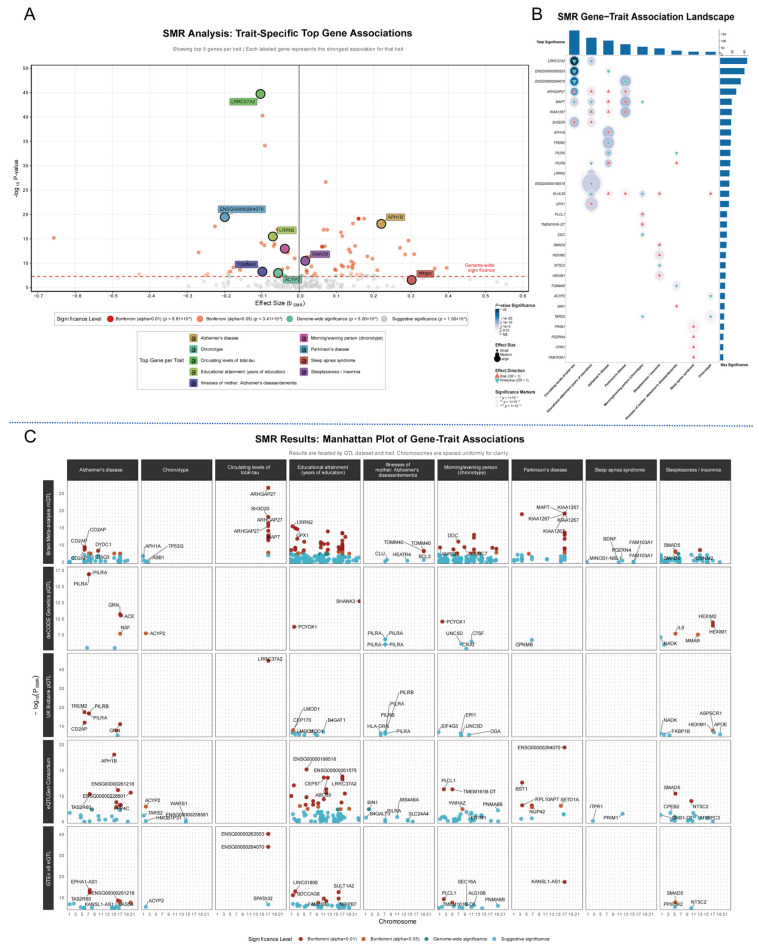
Integrative SMR Analysis Identifies Causal Genes and Pleiotropic Mechanisms Across Sleep and Neurocognitive Traits. (**A**) Volcano plot of trait-specific top gene associations. This plot displays the relationship between SMR effect size (*b*SMR) and statistical significance (−log_10_ *p*-value) for high-confidence gene-trait associations. Each labeled point represents the most significant SMR hit for a given trait, colored by its phenotypic category. The plot highlights several associations that surpass the genome-wide significance threshold (dashed red line), such as the strong association between *APH1B* expression and Alzheimer’s disease (AD) risk (*P*_SMR_ = 8.16 × 10^−19^) and the link between *PCYOX1* protein levels and chronotype (*P*_SMR_ = 1.67 × 10^−9^). (**B**) SMR gene-trait association landscape. This panel provides a comprehensive overview of pleiotropic effects, where genes (y-axis) are associated with multiple traits (x-axis). The size of the circle is proportional to the statistical significance (−log_10_ *P*_SMR_), while the direction of the triangle indicates a risk-increasing (upward) or protective (downward) effect. The bar plot on the right ranks genes by their total significance across all traits, identifying *LRRC37A2* as the top-ranked locus overall, primarily due to its strong association with circulating tau levels. This visualization clearly illustrates the pleiotropic nature of key loci, such as *MAPT*, which is implicated in AD, Parkinson’s disease, and circulating tau, and *PILRA*, which is linked to both AD and its proxy phenotype, “Illnesses of mother: Alzheimer’s disease/dementia.” (**C**) Faceted Manhattan plots of SMR results by trait and QTL dataset. This series of plots dissects the SMR associations to reveal their specific molecular and tissue-of-origin context. Each row represents a different QTL dataset (Brain Meta-analysis mQTL, deCODE pQTL, UKB-PPP pQTL, eQTLGen Consortium, GTEx v8 eQTL), and each column represents a different trait. This breakdown highlights the diverse origins of the signals. For instance, the strong associations for dementia-related traits are driven by multiple omics layers: brain-specific methylation QTLs (*ARHGAP27* with circulating tau, *P*_SMR_ = 2.11 × 10^−27^), protein QTLs (*PILRA* with AD, *P*_SMR_ = 1.18 × 10^−17^), and blood-based eQTLs (*APH1B* with AD, *P*_SMR_ = 8.16 × 10^−19^). Similarly, top signals for sleep-related traits originate from various sources, including pQTLs for Chronotype (*PCYOX1*, *P*_SMR_ = 1.67 × 10^−9^) and eQTLs for Sleeplessness/Insomnia (*SMAD5*, *P*_SMR_ = 3.25 × 10^−11^).

**Figure 4 cimb-47-00967-f004:**
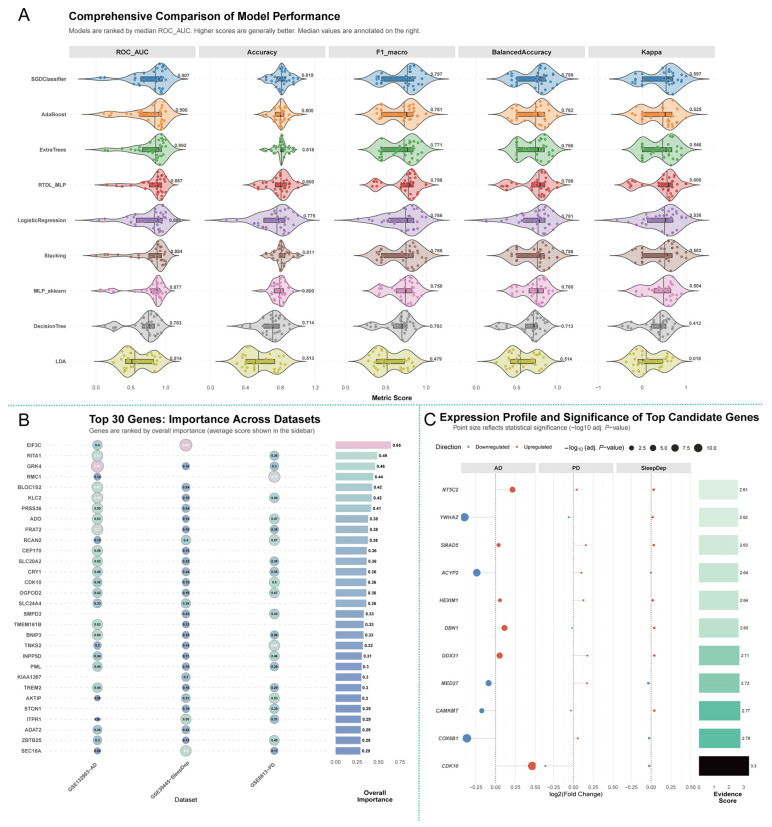
Predictive Modeling and Validation of the Prioritized Transcriptomic Signature. (**A**) Comprehensive comparison of machine learning model performance. This panel displays violin plots comparing the performance of nine different machine learning algorithms across five key metrics (ROC-AUC, Accuracy, F1-macro, Balanced Accuracy, and Kappa). Models were trained to distinguish between disease and control states using the expression levels of prioritized genes. The results show that the deep learning model (RTDL-MLP) and the SGD Classifier consistently outperform other models, with RTDL-MLP achieving the highest mean ROC-AUC of 0.811 and the SGD Classifier showing a median ROC-AUC of 0.907. (**B**) Top 30 genes ranked by importance across datasets. This panel identifies the most influential genes within the predictive models. The bar chart on the right ranks the top 30 genes by their overall importance, calculated as the average normalized score across three independent transcriptomic datasets (Alzheimer’s disease: GSE132903-AD, Sleep: GSE39445-Sleep, Parkinson’s disease: GSE6613-PD). The dot plot on the left shows the dataset-specific importance score for each gene. *EIF3C* is the top-ranked gene overall (average score = 0.65), driven by its high importance in the sleep dataset. In contrast, *GRK4* is the most important gene in the AD dataset (score = 0.91), and *RMC1* is the most important in the PD dataset (score = 0.73). (**C**) Expression profile and significance of top candidate genes. This plot displays the log_2_(Fold Change) of prioritized genes across three conditions: Alzheimer’s disease (AD), Parkinson’s disease (PD), and sleep deprivation (SleepDep). Genes are ranked by a final evidence score, shown in the right-hand bar plot. Point color indicates the direction of regulation (red for upregulated, blue for downregulated), and point size corresponds to the statistical significance (−log_10_ adjusted *p*-value). The analysis reveals a distinct transcriptomic signature in AD, characterized by the significant upregulation of *CDK10* (logFC = 0.46, adj. *p* = 5.85 × 10^−11^) and the downregulation of key genes such as *YWHAZ* (logFC = −0.39, adj. *p* = 1.28 × 10^−10^) and *COX6B1* (logFC = −0.36, adj. *p* = 3.52 × 10^−11^).

**Figure 5 cimb-47-00967-f005:**
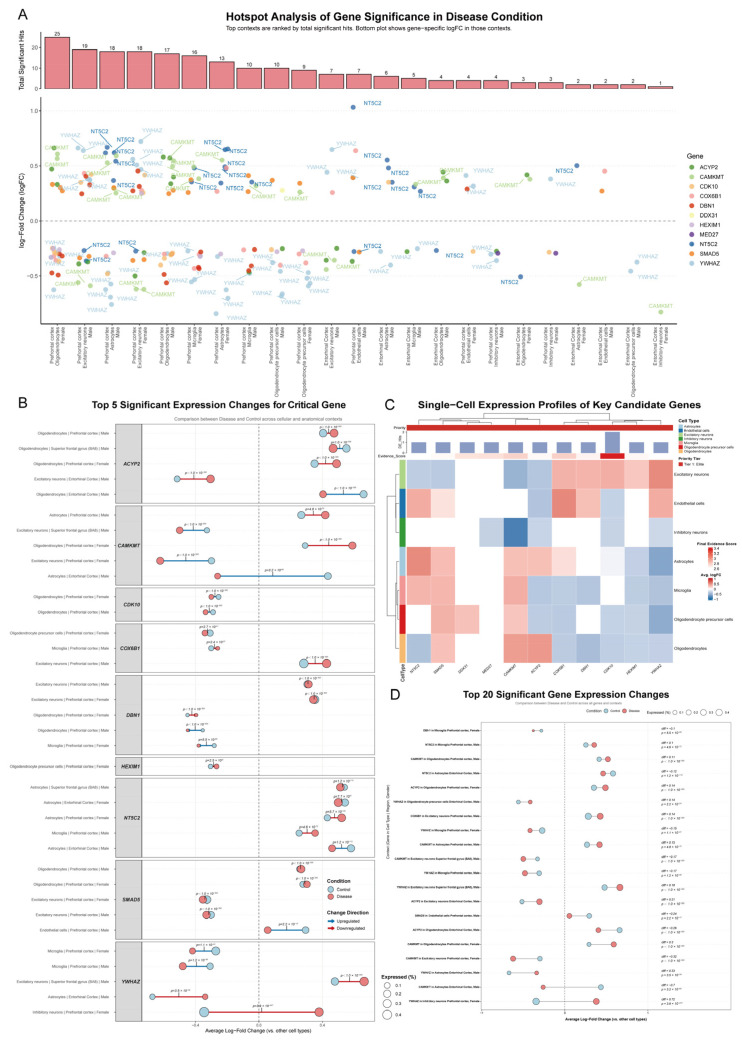
Cell-Type-Specific Expression Profiling and Prioritization of Candidate Genes in Alzheimer’s Disease. (**A**) Hotspot analysis of gene significance in the disease condition. This panel identifies the cellular contexts with the most widespread transcriptional changes. The top bar chart ranks contexts (defined by cell type, brain region, and sex) by the total number of significantly dysregulated candidate genes. Oligodendrocytes from the female prefrontal cortex emerged as the most prominent hotspot, with 25 significant hits, followed by excitatory neurons from the male prefrontal cortex (19 hits) and astrocytes from the male prefrontal cortex (18 hits). The bottom plot shows the log-Fold Change (logFC) of specific genes within these ranked contexts, highlighting cell-type-dependent patterns, such as the consistent downregulation of *YWHAZ* in glial cells (e.g., male prefrontal cortex astrocytes, logFC = −0.76, adj. *p* = 2.60 × 10^−113^) and its upregulation in excitatory neurons (logFC = 0.66, adj. *p* < 1 × 10^−300^). (**B**) Top 5 significant expression changes for critical genes. This panel visualizes the cell-type-specific expression of a selection of top genes, comparing their average logFC in control (blue) versus disease (red) conditions. It reveals complex regulatory shifts, such as the opposing regulation of *CAMKMT*, which is significantly upregulated in disease within female prefrontal cortex oligodendrocytes (logFC increase from 0.29 to 0.59, *p* < 1 × 10^−300^) but is concurrently downregulated in female prefrontal cortex excitatory neurons (logFC decrease from −0.30 to −0.62, *p* < 1 × 10^−300^). (**C**) Single-cell expression profiles of key candidate genes. This heatmap displays the average logFC of the final prioritized genes across major brain cell types in the Alzheimer’s disease brain, revealing distinct co-expression modules. Hierarchical clustering identifies a gene cluster (including *NT5C2*, *SMAD5*, and *DDX31*) characterized by significant upregulation in excitatory neurons (e.g., *NT5C2* Avg. logFC = 0.53). Conversely, a second cluster (*CAMKMT*, *ACYP2*, *CDK10*) exhibits a strong signature of downregulation in inhibitory neurons, with *CAMKMT* showing the most pronounced effect (Avg. logFC = −0.83). The opposing regulation of key genes is evident, with *YWHAZ* being strongly downregulated in glial populations (e.g., Astrocytes, Avg. logFC = −0.58) while being the most upregulated gene in excitatory neurons (Avg. logFC = 0.53). (**D**) Top 20 most significant gene expression changes between disease and control. This dumbbell plot ranks the 20 most significant expression shifts across all genes and contexts by the magnitude of the difference in logFC between disease and control. The most dramatic change is the complete reversal of *YWHAZ* expression in inhibitory neurons of the female prefrontal cortex, which shifted from a repressed state in controls (Avg. logFC = −0.35) to a strongly expressed state in disease (Avg. logFC = 0.38), a massive difference of 0.72 (*p* = 3.9 × 10^−247^). The second-largest shift is the downregulation of *CAMKMT* in astrocytes of the male entorhinal cortex, which changed from high expression in controls to a repressed state in disease (diff = −0.70, *p* = 3.2 × 10^−66^). This highlights profound and often opposing cell-type-specific responses to disease, as also seen for *ACYP2*, which is downregulated in oligodendrocytes (diff = −0.26) but upregulated in excitatory neurons (diff = 0.21).

**Figure 6 cimb-47-00967-f006:**
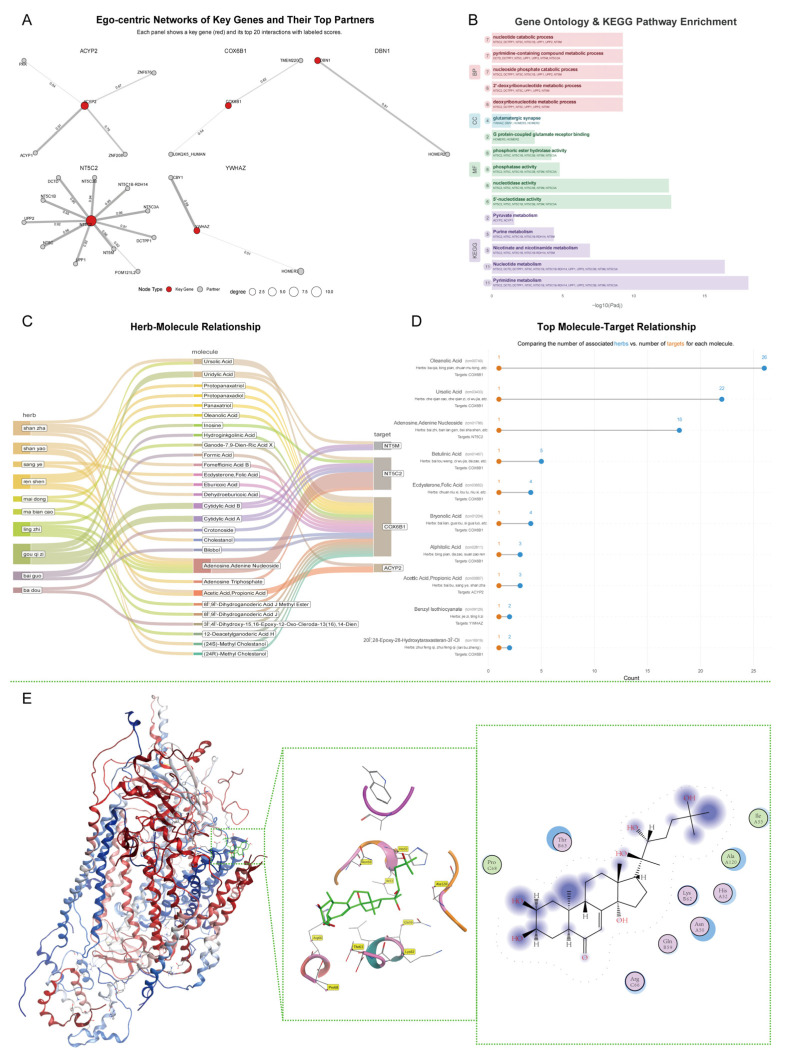
Functional Network Analysis, Therapeutic Target Exploration, and Molecular Validation of Prioritized Genes. (**A**) Ego-centric protein–protein interaction (PPI) networks of key genes. This panel displays the immediate interaction partners for six elite candidate genes. Key genes are shown as red nodes, and their partners are grey nodes. The thickness of the connecting lines (edges) represents the interaction confidence score. The analysis indicates that *NT5C2* is a prominent hub with 11 partners, primarily from the 5′-nucleotidase family (e.g., *NT5C1B*, score = 0.95). In contrast, other genes like *YWHAZ* and *DBN1* show sparser connections but are linked to synaptic scaffolding proteins (*HOMER3* and *HOMER2*, respectively). (**B**) Gene Ontology (GO) and KEGG pathway enrichment analysis. This bar chart shows the top enriched functional terms for the prioritized genes and their interactors. The length of the bars corresponds to the statistical significance (−log_10_ adjusted *p*-value). The analysis highlights two primary functional axes: a highly significant enrichment in “Pyrimidine metabolism” (KEGG, adj. *p* = 8.26 × 10^−19^) and “Nucleotide metabolism” (KEGG, adj. *p* = 3.78 × 10^−17^), driven by the *NT5C2* subnetwork; and a significant enrichment in the “glutamatergic synapse” pathway (GO:CC, adj. *p* = 0.046), driven by *YWHAZ*, *DBN1*, and their *HOMER* partners. (**C**) Herb-molecule-target network. This Sankey diagram illustrates potential relationships between traditional Chinese medicine herbs, their constituent molecules, and our final prioritized gene targets. The selection process was entirely data-driven and gene-centric. First, the six highest-confidence gene targets (right) were chosen based on a convergence of genetic, transcriptomic, and predictive evidence from our preceding analyses. Second, a comprehensive database query was performed to identify all bioactive molecules (middle) with documented interactions against these specific targets. The corresponding herbs displayed (left) are the direct result of this query; they represent the botanical sources of these molecules and were not pre-selected based on any prior hypothesis. The flows highlight that *COX6B1* is the most frequently targeted gene, linked to compounds from numerous herbs like *shan zha* (Hawthorn fruit) and *ren shen* (Ginseng). It also shows that *NT5C2* is consistently targeted by adenosine derivatives found across multiple herbs. (**D**) Top molecule-target relationships. This dumbbell plot ranks bioactive molecules by the number of associated herbs (blue points). It shows that triterpenoids like Oleanolic Acid (from 26 herbs) and Ursolic Acid (from 22 herbs) are the most widespread compounds, primarily targeting *COX6B1*. Adenosine/Adenine Nucleoside (from 18 herbs) is the next most common, exclusively targeting *NT5C2*. (**E**) Molecular docking of Ecdysterone with COX6B1. This panel provides visual validation of the interaction between Ecdysterone and COX6B1. The left image shows the overall protein structure with Ecdysterone (green) docked in the binding pocket. The middle and right images provide detailed 3D and 2D views of the binding site, illustrating that Ecdysterone is stabilized by a network of hydrogen bonds (e.g., with Asn50, Gln59) and hydrophobic interactions (e.g., with His52, Ile53, Lys62), supporting the high-affinity docking score of −5.73 kcal/mol. The protein structure is colored by chain progression from the N-terminus (blue) to the C-terminus (red).

**Table 1 cimb-47-00967-t001:** GWAS Study Profiles.

Id	Trait	Ncase	Ncontrol	Sample_Size	Year	Pmid	Population	Nsnp
**ebi-a-GCST90095138**	Circulating levels of total-tau	__	__	14,721	2022	35396452	European	8,360,926
**ebi-a-GCST90029013**	Educational attainment (years of education)	__	__	461,457	2018	29892013	European	11,972,619
**ebi-a-GCST90027158**	Alzheimer’s disease	39,106	46,828	487,511	2022	35379992	European	20,921,626
**ebi-a-GCST90018916**	Sleep apnea syndrome	13,818	463,035	476,853	2021	34594039	European	24,183,940
**ukb-b-3957**	Sleeplessness/insomnia	__	__	462,341	2018	__	European	9,851,867
**ukb-b-4956**	Morning/evening person (chronotype)	__	__	413,343	2018	__	European	9,851,867
**ukb-b-14699**	Illnesses of mother: Alzheimer’s disease/dementia	36,548	387,190	423,738	2018	__	European	9,851,867
**ukb-a-527**	Diagnoses—main ICD10: G47 Sleep disorders	2025	335,174	337,199	2017	__	European	10,894,596
**ukb-a-210**	Illnesses of mother: Alzheimer’s disease/dementia	26,757	283,086	308,780	2017	__	European	10,894,596
**ieu-b-7**	Parkinson’s disease	33,674	449,056	482,730	2019	__	European	17,891,936
**ieu-a-1087**	Chronotype			128,266	2016	27494321	European	17,032,431
**ebi-a-GCST006685**	Sleep duration (oversleepers)	10,102	81,204	91,306	2016	27494321	European	16,563,303

Note: “__” indicates that information is not available.

**Table 2 cimb-47-00967-t002:** Gene Expression Study Profiles.

Disease/Condition Investigated	GEO Accession	Sample Cohort	Study Synopsis	Publications (PMID)
**Alzheimer’s Disease (AD)**	GSE132903	A total of 195 post-mortem brain tissue samples were analyzed, comprising: AD Cases: 97 Non-demented Controls: 98	This investigation performed transcriptomic profiling on post-mortem middle temporal gyrus (MTG) tissue to elucidate the molecular landscape of Alzheimer’s Disease. Utilizing Illumina microarrays, the study identified a significant number of differentially expressed genes between AD cases and controls. Subsequent Weighted Gene Co-expression Network Analysis (WGCNA) revealed distinct gene modules implicated in key biological pathways, including synaptic function, RNA metabolism, and processes involving the mitochondria-associated membrane (MAM).	Piras et al., 2019 [21]
**Insufficient Sleep/Sleep Restriction**	GSE39445	A cohort of 26 human subjects, from whom a total of 438 blood samples were collected across multiple time-points under varying sleep conditions.	This study was designed to investigate the transcriptomic consequences of insufficient sleep. Employing a crossover experimental design, participants underwent two distinct conditions: one week of sufficient sleep followed by one week of sleep restriction (6 h per night). Following each condition, subjects were subjected to a period of extended wakefulness, during which serial blood samples were collected. Gene expression profiling was performed on RNA isolated from circulating leukocytes using whole-genome microarrays to assess changes in the human blood transcriptome.	Möller-Levet et al., 2013 [23] Laing et al., 2019 [24]
**Parkinson’s Disease (PD)**	GSE6613	A total of 106 individuals, consisting of: Parkinson’s Disease Patients: 50 Patients with Other Neurodegenerative Diseases: 33 Healthy Controls: 23	This study performed a transcriptome-wide analysis of whole blood to discover molecular biomarkers for early-stage Parkinson’s Disease. Gene expression profiles were systematically compared among PD patients, individuals with other neurodegenerative conditions, and healthy controls. The investigation successfully identified and validated a robust gene expression signature that is significantly associated with PD risk, highlighting the utility of peripheral blood as a viable medium for biomarker discovery.	Scherzer et al., 2007 [22] Scherzer et al., 2008 [25]

## Data Availability

The original contributions presented in this study are included in the article/Appendix A. Further inquiries can be directed to the corresponding authors.

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
