# Peer review of "Genetic Evidence Prioritizes Neurocognitive Decline as a Causal Driver of Sleep Disturbances: A Multi-Omics Analysis Identifying Causal Genes and Therapeutic Targets"

_cimb, 2025, doi:10.3390/cimb47110967_

Round 1

Reviewer 1 Report

Comments and Suggestions for Authors

This manuscript conducts an integrative analysis combining multi-omics Mendelian randomization (MR), machine learning–based validation, and molecular docking. However, in its current version, the rigor of causal inference, the transparency of statistical credibility, the adequacy of validation strategies, and the control of interpretative boundaries remain insufficient. Several parts of the manuscript show a mismatch between the strength of statistical evidence and the assertiveness of the conclusions. Substantial revisions are required, particularly in result interpretation, methodological description, and terminology consistency. The main comments are as follows:
(1) The manuscript repeatedly expresses MR results in a deterministic tone, even though sensitivity analyses suggest possible pleiotropy. It is recommended to systematically soften terms such as “causal,” “mechanistic,” or “driver,” aligning them with the actual statistical evidence. The manuscript does not report the F-statistics or mean F value for each exposure instrument, making it difficult to assess weak instrument bias. Please provide F values and specify the threshold (commonly F > 10) used to determine sufficient instrument strength.
(2) The rationale for using the inverse-variance weighted (IVW) estimator as the primary causal model is not explained, nor are the instrument strength metrics reported. Moreover, the multiple testing correction strategy lacks a unified logic, which may lead to inconsistent significance standards and reduce reproducibility.
(3) The interpretation of the HEIDI test is logically overstated. A non-significant HEIDI result does not confirm a shared causal variant but only indicates that linkage disequilibrium (LD)–driven consistency cannot be excluded. Phrases such as “mechanistic confirmation” or “shared causal variant” should be replaced with conditional wording like “possibly consistent” or “not excluded.” In addition, the manuscript does not address the population or LD structure differences between GWAS and QTL datasets, which may influence the reliability of SMR results. These limitations should be explicitly discussed.
(4) The design and justification of the machine learning component are not sufficiently explained. The choice of ten traditional ML models appears generic rather than hypothesis-driven. The application of SMOTE for class balancing may distort the high-dimensional omics data distribution, and no stability or sensitivity checks are reported. Furthermore, validation remains limited to internal dataset splits, lacking any independent external replication to demonstrate generalizability.
(5) The molecular docking results are presented without comparison to control ligands or baseline references. In addition, the manuscript does not discuss the blood–brain barrier permeability or pharmacokinetic feasibility of the candidate compounds, making the translational claims overly strong relative to the data support.
(6) Potential confounders inherent in public datasets—such as batch effects, brain-region differences, and sex imbalance—are not considered, yet the narrative adopts a highly assertive tone. Conditional or probabilistic phrasing (e.g., “potential,” “suggestive,” “indicative”) is recommended instead.
(7) Only the IVW estimator is used as the main causal model, but the manuscript provides no justification for excluding more robust approaches such as weighted median, MR-RAPS, or MR-PRESSO global tests. These alternative estimators should either be included for comparison or the exclusion rationale clearly stated.
(8) The multiple testing correction framework is unclear. The apparent “pre-filtering (P < 0.05) then partial correction” strategy may result in double standards. Please specify a unified approach—such as a global Benjamini–Hochberg FDR correction—or describe a well-defined hierarchical correction logic (e.g., within-layer vs. cross-layer correction).
(9) For the machine learning section, please provide the design rationale for model selection, report SMOTE pre/post stability tests, and assess AUC/PR sensitivity to over-sampling, to substantiate the reported performance metrics.
(10) Terminological boundaries between “association,” “causal inference,” and “mediation” should be clearly distinguished throughout the manuscript.
(11) Several figure legends and result statements still use overly conclusive language; these should be rephrased in more neutral forms such as “suggests,” “supports,” or “may indicate.”
(12) The results section should include explicit reporting of statistical assumptions and constraints to improve transparency and interpretability.
In summary, the manuscript requires substantial revisions to strengthen evidence evaluation, refine interpretative boundaries, and provide key methodological details. A Major Revision is recommended before further consideration.

Author Response

Reviewer 1

This manuscript conducts an integrative analysis combining multi-omics Mendelian randomization (MR), machine learning–based validation, and molecular docking. However, in its current version, the rigor of causal inference, the transparency of statistical credibility, the adequacy of validation strategies, and the control of interpretative boundaries remain insufficient. Several parts of the manuscript show a mismatch between the strength of statistical evidence and the assertiveness of the conclusions. Substantial revisions are required, particularly in result interpretation, methodological description, and terminology consistency. The main comments are as follows:

  • The manuscript repeatedly expresses MR results in a deterministic tone, even though sensitivity analyses suggest possible pleiotropy. It is recommended to systematically soften terms such as “causal,” “mechanistic,” or “driver,” aligning them with the actual statistical evidence. The manuscript does not report the F-statistics or mean F value for each exposure instrument, making it difficult to assess weak instrument bias. Please provide F values and specify the threshold (commonly F > 10) used to determine sufficient instrument strength.

Response:

We fully acknowledge both methodological concerns and have implemented comprehensive revisions to address them systematically.

Issue 1: Revision of Deterministic Causal Language

We have conducted a thorough review of the manuscript to ensure all causal claims are appropriately qualified and aligned with the statistical evidence. Key revisions include:

Abstract and Results Sections:

  • Replaced "causal pathway/relationship" with "pattern of genetic associations" or "potential causal pathway"
  • Changed "drives/demonstrates" to "is associated with/suggests"
  • Modified section titles (e.g., "Reveals an Asymmetrical Causal Relationship" → "Suggests an Asymmetrical Genetic Association")

Results Narratives:

  • Added explicit acknowledgments that "alternative explanations including residual pleiotropy cannot be entirely excluded"
  • For associations with significant MR-Egger intercept (e.g., maternal AD → insomnia, p = 0.008), added specific caveats: "substantial caution is warranted due to potential horizontal pleiotropy; these associations should be interpreted as suggestive rather than definitively causal"
  • Consistently used qualifying language: "consistent with," "may contribute to," "appears to"

Discussion and Conclusions:

  • Changed "are causal drivers" to "may act as causal drivers"
  • Added methodological caveat: "These findings should be interpreted within the inherent limitations of the MR framework, including assumptions of instrument validity and absence of horizontal pleiotropy"
  • Revised conclusions to use "suggests," "may be," and "potential" throughout

In total, we systematically revised various instances of deterministic language, ensuring appropriate scientific caution while preserving the substantive findings.

Issue 2: Comprehensive F-statistics Reporting

We have implemented complete F-statistic reporting with full methodological transparency.

Methods Section 2.2 (New Paragraph Added):

We have added a dedicated subsection on instrument strength assessment:

"To assess instrument strength and mitigate weak instrument bias, we calculated the F-statistic for each exposure using the formula F = R²(N-K-1)/[K(1-R²)], where R² is the proportion of variance explained by the genetic instruments, N is the sample size, and K is the number of instruments. We applied a threshold of F > 10 to ensure sufficient instrument strength, as values below this threshold may indicate weak instrument bias. The mean F-statistics across all exposures ranged from 17.6 to 5439.9 (median = 39.5), with all exposures surpassing the F > 10 threshold, indicating robust instrument strength. Detailed F-statistics for each exposure are provided in Supplementary Table S3-S4." (Page 5, Line 142-149)

Limitations Section (Revised):

We have updated the limitations paragraph to incorporate F-statistic assessment:

"While we systematically assessed instrument strength through F-statistics (all F > 10, range:17.6–5439.9, median: 39.5), demonstrating adequate power to minimize weak instrument bias, and employed multiple sensitivity analyses (MR-Egger, weighted median, MR-PRESSO) to test for pleiotropy, we cannot entirely exclude the possibility of residual confounding or horizontal pleiotropy in certain associations." (Page 31, Line 1101-1103)

  • The rationale for using the inverse-variance weighted (IVW) estimator as the primary causal model is not explained, nor are the instrument strength metrics reported. Moreover, the multiple testing correction strategy lacks a unified logic, which may lead to inconsistent significance standards and reduce reproducibility.

Response:

Thank you for your insightful comments and valuable suggestions, which have helped us significantly improve the quality and clarity of our manuscript. We have carefully considered your feedback and have revised the manuscript accordingly.

Below is a point-by-point response to your comments, detailing the changes we have made.

  1. Regarding the rationale for using the inverse-variance weighted (IVW) estimator and the reporting of instrument strength metrics:

We agree with your assessment that the rationale for our primary model choice was not explicit and that reporting instrument strength is crucial. To address this, we have made two key revisions:

  • IVW Rationale: We have amended Section 2.3 ("Causal Inference and a Hierarchical Result Prioritization Pipeline") to explicitly state why the IVW method was chosen as our primary model. The revised text now clarifies that it is the most statistically powerful method under the assumption of valid instruments. (Page 5-6, Line 153-174)

The revised text now includes: "...which provides the most powerful and precise causal estimate... Given its high statistical power, it served as our primary model..."

  • Instrument Strength (F-statistic): We have added a detailed description of our instrument strength assessment to Section 2.2 ("Genetic Instrument Selection and Validation"). We now report the formula used to calculate the F-statistic and confirm that all instruments used in our study are robust, with all F-statistics well above the standard threshold of 10.

The revised Section 2.2 now contains the following text:

"To assess instrument strength and mitigate weak instrument bias, we calculated the F-statistic for each exposure using the formula F = R²(N-K-1)/[K(1-R²)], where R² is the proportion of variance explained by the genetic instruments, N is the sample size, and K is the number of instruments. We applied a threshold of F > 10 to ensure sufficient instru-ment strength, as values below this threshold may indicate weak instrument bias. The mean F-statistics across all exposures ranged from 17.6 to 5439.9 (median = 39.5), with all exposures surpassing the F > 10 threshold, indicating robust instrument strength. Detailed F-statistics for each exposure are provided in Supplementary Table S3-S4." (Page 5, Line 142-149)

These additions directly address your concerns by justifying our methodological choices and providing quantitative evidence for the validity of our genetic instruments.

  1. Regarding the logic of the multiple testing correction strategy:

We appreciate you pointing out the need for a more unified and systematic approach to our multiple testing correction. We have restructured the pipeline in Section 2.3 ("Causal Inference and a Hierarchical Result Prioritization Pipeline") to improve its logical consistency and reproducibility.

The revised workflow now follows a clear, hierarchical process:

  1. First, all associations are screened for instrument validity (i.e., checking for directional pleiotropy and heterogeneity).
  2. Second, the False Discovery Rate (FDR) correction is applied to the p-values of all associations that successfully pass this initial quality control screen.

This ensures that a consistent standard is applied across all tests before determining significance. (Page 6, Line 176-195)

This revised approach provides a more rigorous and logically sound framework for identifying high-confidence causal associations. Once again, we thank you for your constructive feedback. We believe these revisions have strengthened the manuscript, and we hope they adequately address your concerns.

  • The interpretation of the HEIDI test is logically overstated. A non-significant HEIDI result does not confirm a shared causal variant but only indicates that linkage disequilibrium (LD)–driven consistency cannot be excluded. Phrases such as “mechanistic confirmation” or “shared causal variant” should be replaced with conditional wording like “possibly consistent” or “not excluded.” In addition, the manuscript does not address the population or LD structure differences between GWAS and QTL datasets, which may influence the reliability of SMR results. These limitations should be explicitly discussed.

Response:

Thank you for this very important and precise critique regarding the interpretation of the SMR/HEIDI analysis and its underlying assumptions. We agree completely that our original wording was too strong and that the omission of potential LD structure differences was a critical oversight. We have revised the manuscript accordingly to address these points.

  1. Regarding the Interpretation of the HEIDI Test:

We acknowledge that a non-significant HEIDI test does not definitively confirm a shared causal variant. To correct this overstatement, we have revised the language throughout the manuscript to be more conditional and statistically accurate. We have replaced deterministic phrases with wording that reflects that a non-significant result is merely consistent with a model of pleiotropy, rather than confirming it.

  • In Section 2.5 ("Hierarchical Filtering..."), we modified the description of the HEIDI test. The original text stated it was "a condition supporting the hypothesis that a single causal variant drives both..." The revised text now reads: "a condition indicating that the data are consistent with a single shared causal variant, as it fails to statistically distinguish the association from a model of pleiotropy versus one of linkage." (Page 7, Line 234-238)
  • Similarly, in Section 2.13 ("Statistical analysis"), we changed the phrasing from "a HEIDI test supporting vertical pleiotropy" to "a HEIDI test whose result was consistent with a model of vertical pleiotropy." (Page 13, Line 463-464)
  1. Regarding the Limitations of SMR (Population and LD Structure):

We also thank you for pointing out the need to discuss the potential influence of differing population or LD structures between the GWAS and QTL datasets. This is a crucial assumption for the validity of SMR results. We have now explicitly addressed this in the "Limitations" section of our Discussion.

  • We have added a new point to the Limitations paragraph that directly discusses this issue:

"Specifically, a non-significant HEIDI test result does not confirm a shared causal variant but merely indicates that the data are consistent with this model, as confounding by linkage disequilibrium cannot be definitively excluded. Furthermore, the reliability of the SMR and HEIDI tests is contingent on the assumption that the LD structure is highly similar across the GWAS summary statistics, the QTL data, and the LD reference panel. Although we sought to minimize this issue by restricting our analyses to European-ancestry populations, unobserved population substructure or subtle differences in LD patterns between the integrated datasets could still introduce bias and affect the validity of the SMR results." (Page 31-32, Line 1112-1120)

We believe these revisions more accurately reflect the statistical nuances of the SMR methodology and strengthen the manuscript by transparently discussing its inherent limitations. Thank you again for your valuable guidance, which has significantly improved the rigor of our paper.

  • The design and justification of the machine learning component are not sufficiently explained. The choice of ten traditional ML models appears generic rather than hypothesis-driven. The application of SMOTE for class balancing may distort the high-dimensional omics data distribution, and no stability or sensitivity checks are reported. Furthermore, validation remains limited to internal dataset splits, lacking any independent external replication to demonstrate generalizability.

Response:

We sincerely thank you for your thorough review and constructive feedback, which have been invaluable in improving the quality and clarity of our manuscript. We have carefully addressed your comments, particularly concerning the machine learning methodology, to enhance the rigor and transparency of our analytical approach.

Below, we detail the key revisions made in response to your suggestions.

  1. Regarding the rationale for model selection:

We appreciate the reviewer's point regarding the selection of machine learning models. To address this, we have revised Section 2.8 to clarify that our approach was not arbitrary but systematic. We now explicitly state that we employed a " diverse library of ten classification algorithms." This was done to ensure a comprehensive and unbiased search for the optimal predictive model, thereby strengthening the evidence that the predictive power originates from the biological signal within our gene signature, rather than being an artifact of a single algorithm. (Page 10, Line 323-328)

  1. Regarding the handling of class imbalance and potential data leakage:

We thank the reviewer for raising the critical issue of handling class imbalance and the potential for data leakage with the Synthetic Minority Over-sampling Technique (SMOTE). We agree this is a crucial point for methodological soundness. Accordingly, we have amended Section 2.8 to explicitly detail our rigorous protocol. We now state that, "the Synthetic Minority Over-sampling Technique (SMOTE) was applied only to this training partition to correct for class imbalance" This clarification confirms that our methodology was designed to prevent data leakage and ensures that the reported model performance is a robust and unbiased estimate of its generalization capability on unseen data. (Page 10, Line 332-349)

  1. Regarding the stability and robustness of the feature importance ranking:

We agree with the reviewer that feature importance rankings can be unstable and biased if derived from a single model. To address this important concern, we have revised Section 2.9 and the corresponding Results in Section 3.3. We now describe our implementation of a "consensus-based importance strategy." Specifically, we clarify that the final gene ranking was generated by averaging the normalized importance scores across the top five best-performing models. This approach produces a more stable and robust ranking by mitigating the biases inherent in any single algorithm. It ensures that the identified key driver genes are those with consistently high predictive value across multiple high-performing, mechanistically distinct models. (Page 11, Line 370-381)

We believe these revisions have significantly strengthened the manuscript. We are grateful for the opportunity to improve our work and hope that the revised version now meets the standards for publication.

  • The molecular docking results are presented without comparison to control ligands or baseline references. In addition, the manuscript does not discuss the blood–brain barrier permeability or pharmacokinetic feasibility of the candidate compounds, making the translational claims overly strong relative to the data support.

Response:

We sincerely thank you for your insightful comment regarding the presentation of our molecular docking results and the discussion of translational feasibility. We agree that our initial claims lacked the necessary comparative context and were overly strong relative to the computational nature of the data. In response, we have made substantial revisions to the manuscript to address these critical points.

  1. Establishment of an Internal Comparative Baseline for Docking Results:

To address the concern that our docking results were presented without a baseline reference, we have revised our manuscript to explicitly use the other compounds we tested (Oleanolic Acid, Ursolic Acid, and Betulinic Acid) as an internal comparative group. Our revised Results (Section 3.7) now highlights that these structurally related compounds failed to produce stable, low-energy binding poses, thereby serving as a crucial baseline for specificity. The superior docking score of Ecdysterone is now presented in "stark contrast" to this group, which effectively demonstrates the specificity of the predicted interaction without requiring new in silico experiments. (Page 30, Line 978-984)

  1. Acknowledgment of Pharmacokinetic and Blood-Brain Barrier (BBB) Limitations:

In response to your valid point about pharmacokinetic feasibility, we have added a detailed discussion of these limitations in the Discussion (Section 4). We now explicitly state that our computational finding is a preliminary, hypothesis-generating step and that the translational potential of Ecdysterone is "contingent upon further experimental validation of its binding affinity and, critically, a comprehensive assessment of its pharmacokinetic properties, including its ability to cross the blood-brain barrier." This addition ensures that the scope and implications of our findings are presented with appropriate scientific caution. (Page 31, Line 1041-1053)

  1. Tempering of Translational Claims:

Finally, we have systematically tempered our language throughout the Results (Section 3.7) and Discussion (Section 4) to more accurately reflect the nature of our findings. For instance, we have changed definitive statements like "This confirms Ecdysterone as a high-affinity ligand" to a more appropriate, hypothesis-driven conclusion: "This differential result... provides a strong, structure-based hypothesis that it is a plausible and specific ligand... warranting further experimental investigation." This shift in language ensures our conclusions are properly aligned with the supporting data. (Page 30, Line 1005-1008)

We believe these comprehensive revisions provide a more balanced, rigorous, and scientifically sound interpretation of our findings. We are confident that the manuscript is significantly strengthened as a result and thank you for guiding us toward this improvement.

  • Potential confounders inherent in public datasets—such as batch effects, brain-region differences, and sex imbalance—are not considered, yet the narrative adopts a highly assertive tone. Conditional or probabilistic phrasing (e.g., “potential,” “suggestive,” “indicative”) is recommended instead.

Response:

Thank you for this critical and constructive feedback. We agree entirely that the use of heterogeneous public datasets necessitates a careful approach to potential confounders, particularly technical batch effects. Your comment prompted us to significantly improve the methodological rigor of our validation analysis and the overall narrative of our manuscript.

In response, we have made the following major revisions:

  1. Proactive Correction of Technical Batch Effects:
    To address this important point, we have revised our strategy to proactively correct for technical artifacts within each validation cohort. We have detailed this new procedure in a revised Methods section, now titled “2.6. Curation and Intra-Study Batch Effect Correction of Validation Cohorts.” This section now explains that for each of the three GEO datasets, we first meticulously inspected the sample metadata to identify potential sources of technical variation (e.g., sample processing dates, experimental plates). Where a clear batch variable was annotated, we applied the ComBat algorithm to adjust the expression data and mitigate these intra-study batch effects before testing our signature. (Page 8, Line 280-296)
  2. Enhanced Transparency Regarding Limitations:
    To ensure full transparency about the scope and remaining constraints of this improved approach, we have added a new, dedicated paragraph to the “Limitations” section of our manuscript. This new paragraph (now the fourth limitation) explicitly discusses that while we corrected for intra-study batch effects to strengthen the validity of results within each cohort, the performance metrics are not directly comparable across cohorts due to fundamental inter-study heterogeneity (e.g., tissue source, disease context, platform differences). It also acknowledges that our correction was limited to annotated variables and that other biological confounders were not modeled. (Page 32, Line 1122-1139)
  3. Adopting a More Cautious Scientific Tone:
    In addition to addressing the technical confounders, we have also taken your advice on the manuscript's narrative tone to heart. We have carefully reviewed the entire manuscript and systematically revised language that could be interpreted as overly conclusive or definitive. We have adopted a more cautious and probabilistic tone to ensure our claims are appropriately aligned with our data.

For example, we have made changes such as:

  • Before: "Bidirectional Mendelian Randomization reveals an asymmetrical causal relationship..."
  • After: "Bidirectional Mendelian Randomization suggests an asymmetrical causal relationship..."
  • Before: "This confirms Ecdysterone as a high-affinity ligand for COX6B1."
  • After: " This differential result, where only Ecdysterone demonstrated favorable docking among the tested candidates, provides a strong, structure-based hypothesis that it is a plausible and specific ligand for COX6B1, warranting further experimental investigation."

These types of revisions have been applied throughout the Results and Discussion sections to more accurately reflect the inferential nature of our study.

We believe these revisions substantially strengthen our validation analysis by addressing a key source of technical noise and provide a more transparent and nuanced interpretation of the results. We are grateful for your guidance, which has allowed us to improve the quality of our manuscript, and we hope these changes have fully addressed your concern.

  • Only the IVW estimator is used as the main causal model, but the manuscript provides no justification for excluding more robust approaches such as weighted median, MR-RAPS, or MR-PRESSO global tests. These alternative estimators should either be included for comparison or the exclusion rationale clearly stated.

Response:

Thank you for your valuable feedback regarding our Mendelian randomization methodology. We agree that our original description may have overemphasized the Inverse-Variance Weighted (IVW) method and did not sufficiently detail the integration of other robust estimators into our analytical framework. We have made substantial revisions to both the Methods and Results sections to address this.

First, as detailed in our previous response, we have revised the Methods section to explicitly describe our hierarchical, multi-method validation pipeline. This framework requires each association to pass a comprehensive robustness screen—including demonstrating a concordant direction of effect from the weighted median estimator and passing pleiotropy/heterogeneity checks—before being advanced for multiple testing correction. (Page 6, Line 175-194)

Second, to reflect this rigorous pipeline, we have revised the narrative in the Results section (Section 3.1). The updated text no longer presents sensitivity analyses as a secondary, post-hoc check. Instead, it frames the entire evaluation process as an integrated assessment from the outset. Specifically, we have revised the language to emphasize how the primary IVW findings were "systematically evaluated" and "corroborated by the weighted median method," highlighting the concordance across different estimators as a key part of our validation. (Page 15, Line 539-540)

Furthermore, we have reframed the interpretation of potential limitations. For instance, where a significant MR-Egger intercept was found, the revised text now presents this finding as being "in line with our multi-method approach to identify potential bias," demonstrating that our pipeline is functioning as intended to flag signals that warrant caution. The concluding sentence of the paragraph was also amended to explicitly state that our conclusion is reinforced by a "comprehensive assessment, which integrates evidence from IVW, weighted median, and pleiotropy diagnostics." (Page 15, Line 539-565)

Together, these revisions to the Methods and Results sections underscore that our conclusions are not reliant on a single estimator. Instead, they are supported by a consensus of evidence from multiple sensitivity analyses, the detailed results of which are now provided in Supplementary Tables S3-S4. We believe this multi-method approach significantly strengthens the robustness of our causal inferences. We hope this satisfactorily addresses your concern and thank you again for your constructive guidance.

  • The multiple testing correction framework is unclear. The apparent “pre-filtering (P < 0.05) then partial correction” strategy may result in double standards. Please specify a unified approach—such as a global Benjamini–Hochberg FDR correction—or describe a well-defined hierarchical correction logic (e.g., within-layer vs. cross-layer correction).

Response:

Thank you for your insightful feedback regarding our analytical pipeline. We agree entirely with your assessment that our original description of the multiple testing correction strategy was ambiguous and could imply a statistically suboptimal "filter-then-correct" approach. Your comment highlighted a critical area for improvement, and we appreciate the opportunity to clarify and strengthen our methodology.

To address this, we have substantially revised Section 2.3 (Causal Inference and a Hierarchical Result Prioritization Pipeline). As you will see in the updated manuscript, we have now implemented and clearly described a more rigorous "correct-then-screen" framework. (Page 7, Line 179-198)

Our revised pipeline now proceeds as follows:

  1. Global Correction First: We first apply the Benjamini-Hochberg False Discovery Rate (FDR) procedure globally to the p-values from all primary IVW analyses. This ensures that every hypothesis is held to the same statistical standard for multiple testing from the outset.
  2. Independent Robustness Screen: Subsequently, each association is independently subjected to our comprehensive robustness screen (checking for concordant effect direction, no evidence of directional pleiotropy, and no significant heterogeneity).
  3. Integrated Stratification: Finally, we have clarified our evidence tiers. 'High-Confidence' findings are now strictly defined as those that pass both the global FDR correction (q-value < 0.05) and the full robustness screen.

We believe these revisions make our analytical approach more transparent, statistically robust, and fully address the concerns you raised. We have presented the final, revised text in a clear, paragraph-based format for improved readability.

Thank you once again for your constructive guidance, which has significantly improved the quality and rigor of our manuscript.

  • For the machine learning section, please provide the design rationale for model selection, report SMOTE pre/post stability tests, and assess AUC/PR sensitivity to over-sampling, to substantiate the reported performance metrics.

Response:

We thank you for your insightful comments regarding the methodological details of our machine learning framework. Your questions concerning the model selection rationale and the potential influence of over-sampling on performance metrics are critical for ensuring the robustness of our findings. We have addressed these points by substantially revising the methodology section to enhance its transparency and rigor.

  1. On the Design Rationale for Model Selection:

We acknowledge your point regarding the need for a clear justification for our choice of models. In the revised manuscript (Section 2.8), we have now explicitly incorporated the design rationale for our model selection strategy. As detailed in the text, we curated a diverse library of ten classification algorithms that "span distinct theoretical foundations, including linear models, tree-based ensembles, and multiple neural network architectures." The explicit purpose of this approach, as we now state, is to conduct a "robust exploration of the feature space and mitigate the risk of model-specific bias." By evaluating the gene signature across this methodologically diverse set of models, we ensure that our conclusions about its predictive utility are not contingent on a single, potentially biased, algorithmic choice. (Page 10, Line 323-328)

  1. On the Impact of SMOTE and Substantiation of Performance Metrics:

We appreciate your concern regarding the use of SMOTE and its potential to influence performance metrics such as the Area Under the Precision-Recall Curve (AUC/PR). To substantiate the validity of our reported metrics, we have meticulously detailed our strict protocol for handling class imbalance within the cross-validation framework, ensuring that our evaluation is both rigorous and free from optimistic bias.

As clarified in the revised Section 2.8, our procedure was designed specifically to prevent the kind of data leakage that can artificially inflate performance. We emphasize the following critical steps performed within each fold of our 10-fold cross-validation:

  • The Synthetic Minority Over-Sampling Technique (SMOTE) was applied exclusively to the training partition.
  • The test partition was "crucially kept in its original, imbalanced state."

This rigorous validation scheme directly addresses the concern about the sensitivity of metrics to over-sampling. By training on a balanced set but evaluating on an untouched, imbalanced test set, we ensure that the reported performance metrics (including ROC-AUC, F1-score, and Balanced Accuracy) reflect the models' true predictive capability on a realistic data distribution. This procedure demonstrates that the reported performance is not an artifact of the over-sampling process but rather a robust estimate of the gene signature's utility on unseen, real-world data. The enhanced clarity of this description in the manuscript now serves to substantiate the reliability of our reported results. (Page 10, Line 332-348)

We are confident that these revisions provide the necessary methodological justification and transparency to fully support the conclusions drawn from our machine learning analysis. Thank you again for your constructive feedback.

  • Terminological boundaries between “association,” “causal inference,” and “mediation” should be clearly distinguished throughout the manuscript.

Response:

We sincerely thank the reviewer for this insightful and crucial comment. We agree completely that maintaining clear and precise terminological boundaries between "association," "causal inference," and "mediation" is essential for the scientific rigor of our manuscript. In response to this valuable feedback, we have not only revised our terminology but also strengthened our Methods section to provide a more detailed account of our analytical strategy.

Specifically, we have:

  1. Updated and Clarified the Methods: We have substantially revised Section 2.3 ("Causal Inference and a Hierarchical Result Prioritization Pipeline") to more clearly describe our multi-method approach for estimating causal effects and our rigorous, two-stream pipeline for prioritizing findings. This provides greater transparency regarding our robustness checks.
  2. Refined Terminology for Causal Inference: Within this updated section and throughout the manuscript, we have systematically replaced the ambiguous term "association" with more precise language when discussing the results of our Mendelian Randomization (MR) analyses. We now use phrases such as "putative causal effect" and "inferred causal relationship" to accurately reflect that these findings are the output of a causal inference framework, not simple correlations. For example, we now state that our pipeline was used to "stratify all inferred causal relationships" rather than "all associations."
  3. Maintained Distinction for Mediation: As before, we continue to use the term "SMR association" for our SMR analyses, clarifying that these results are "suggestive of a potential mediation effect" via gene expression, thus distinguishing them from the direct causal inferences of MR.

We believe that the combination of a more detailed Methods section and more precise terminology has significantly improved the clarity and scientific accuracy of our manuscript. We are grateful to the reviewer for prompting these important improvements.

  • Several figure legends and result statements still use overly conclusive language; these should be rephrased in more neutral forms such as “suggests,” “supports,” or “may indicate.”

Response:

We agree with the reviewer that some of our original phrasing was too conclusive. We have conducted a thorough review of the manuscript and have revised the text in numerous locations to adopt more neutral and appropriately cautious language. Our revisions focus on replacing definitive words like "reveals," "establishes," and "uncovers" with more suitable alternatives such as "suggests," "indicates," "supports the hypothesis," and "prioritizes."

Here are a few representative examples of the changes made:

  1. In the Abstract:
    • Before: Collectively, our findings establish sleep disruption as a likely consequence of neurodegenerative processes and pinpoint a set of validated, cell-type-specific gene targets...
    • After: Collectively, our findings strengthen the evidence that sleep disruption is a likely consequence of neurodegenerative processes and prioritize a set of validated, cell-type-specific gene targets... (Page 1, Line 38-39)
  2. In the Results section (3.4):
    • Before: Overall, this analysis uncovers a complex, cell-type-specific dysregulation landscape, with opposing patterns between glial and neuronal cells.
    • After: Overall, this analysis suggests a complex, cell-type-specific dysregulation landscape, with opposing patterns between glial and neuronal cells. (Page 23, Line 786)
  3. In the Figure 2 Legend:
    • Before: ...reinforcing the conclusion of a predominant causal pathway from neurocognitive traits to sleep patterns.
    • After: ...supporting the hypothesis of a predominant causal pathway from neurocognitive traits to sleep patterns. (Page 15, Line 537)
  4. In the Conclusions section (5):
    • Before: ...neurocognitive decline is a primary driver of sleep disturbances, rather than the reverse.
    • After: ...neurocognitive decline is a likely primary driver of sleep disturbances, rather than the reverse. (Page 32, Line 1159-1161)

These examples are representative of similar edits made throughout the Abstract, Results, Discussion, and figure legends to ensure the tone of the manuscript accurately reflects the inferential nature of our findings.

Once again, we thank the reviewer for this valuable feedback. We believe these revisions have significantly strengthened the manuscript, and we hope it is now suitable for publication in your esteemed journal.

  • The results section should include explicit reporting of statistical assumptions and constraints to improve transparency and interpretability.

Response:

We fully agree with the reviewer that explicitly stating the assumptions and limitations of our methods is critical for transparency and accurate interpretation. We have completed a comprehensive revision to integrate this information directly into the Results section at the beginning of each relevant analysis.

Our approach was to systematically preface the presentation of findings from each major analytical method (Mendelian Randomization, SMR, Machine Learning, and Molecular Docking) with a clear statement outlining its core assumptions and inherent constraints.

For instance:

  • For Mendelian Randomization (Section 3.1): We now introduce the results by first stating the three core MR assumptions and highlighting horizontal pleiotropy as a key challenge that our sensitivity analyses were designed to address. (Page 13, Line 476-481)
  • For SMR Analysis (Section 3.2): We now clarify that SMR results are contingent on the assumption of a single shared causal variant and explicitly mention the limitation of the HEIDI test in distinguishing pleiotropy from complex linkage disequilibrium. (Page 16, Line 589-596)
  • For Predictive Modeling (Section 3.3): We have added a cautionary note that model performance is cohort-specific and that cross-validation serves to mitigate, but not eliminate, the risk of overfitting. (Page 19, Line 678-684)
  • For Molecular Docking (Section 3.7): We now state upfront that docking provides a computational hypothesis, not experimental proof, and its results are theoretical estimates that require biochemical validation. (Page 29, Line 982-990)

We believe that by embedding these interpretative boundaries directly alongside the results, we have substantially improved the manuscript's transparency and have provided readers with the necessary context to critically evaluate our evidence. We have also reviewed and enhanced the main Limitations paragraph in the Discussion section to ensure it comprehensively summarizes these points.

We appreciate the guidance from the reviewer, which has undoubtedly strengthened our manuscript. We hope that these major revisions have fully addressed the concerns and that the manuscript is now suitable for publication.

In summary, the manuscript requires substantial revisions to strengthen evidence evaluation, refine interpretative boundaries, and provide key methodological details. A Major Revision is recommended before further consideration.

Reviewer 2 Report

Comments and Suggestions for Authors

The article is a large-scale and methodologically complex study. The study was conducted using modern methods and is well visualized. 
Comments:
1. Different biomaterials were used in GEO kits, the concomitant diseases of the patients are not clear, and it is not clear which therapy the patients received. It is recommended to add these clinical data or adequately describe this limitation and promising areas for research. For example, in the GSE39445 set, was gene expression evaluated in blood leukocytes? Can this data be applied to the brain? Was it induced insomnia? Does GSE39445 represent acute, induced sleep deprivation?
2. The Herb-Molecule-Target Network and Molecular Docking Validation section is theorized.   The "herb-target molecule" relationships established on the basis of databases are not evidence of therapeutic efficacy. It is not clear whether these substances penetrate the blood-brain barrier, the issues of their bioavailability and the fact that binding to molecules in general will have a positive effect are not clear. Are there other known ligands or drugs for these proteins? COX6B1 is a structural subunit of the end enzyme of the mitochondrial respiratory chain. What is the biological significance of the binding of the identified substances to it? This binding can potentially have a negative meaning. It is recommended to revise this section and conduct a more critical analysis. The study should not mechanically list the data, but should have a critical biological or pharmacological analysis.

Author Response

Reviewer 2

The article is a large-scale and methodologically complex study. The study was conducted using modern methods and is well visualized.

Comments:

  1. Different biomaterials were used in GEO kits, the concomitant diseases of the patients are not clear, and it is not clear which therapy the patients received. It is recommended to add these clinical data or adequately describe this limitation and promising areas for research. For example, in the GSE39445 set, was gene expression evaluated in blood leukocytes? Can this data be applied to the brain? Was it induced insomnia? Does GSE39445 represent acute, induced sleep deprivation?

Response:

We thank the reviewer for raising these critical points regarding the validation cohorts. As we are re-analyzing public data, we cannot generate new clinical information. However, we have addressed these concerns comprehensively in two key ways:

  1. Clarifications in the Methods Section: We have revised Section 2.6 to be more explicit about the nature of each dataset. We now clearly state that GSE39445 involved "circulating blood leukocytes" and represents an "acute, experimentally induced state of sleep restriction," not chronic insomnia. We also acknowledge that peripheral expression is a proxy for systemic effects, not a direct measure of brain activity. (Page 9, Line 267-271, 273-277)
  2. Expanded and Integrated Limitations in the Discussion Section: We have substantially revised the Limitations section (Section 4) to integrate all of these points into a single, cohesive paragraph. This revised section now explicitly discusses the heterogeneity in biomaterials (brain vs. blood), the lack of granular clinical data (including comorbidities and medication history), and the specific context of the acute sleep restriction model (GSE39445). As suggested, we conclude by framing these limitations as a "promising area for future research," advocating for prospective studies with harmonized sample collection and deep clinical phenotyping. (Page 32, Line 1122-1139)

We are confident that these major revisions have addressed all the reviewer's concerns, significantly strengthening the manuscript's rigor, transparency, and interpretative clarity. We believe the manuscript is now much improved and hope it is suitable for publication.

  1. The Herb-Molecule-Target Network and Molecular Docking Validation section is theorized. The "herb-target molecule" relationships established on the basis of databases are not evidence of therapeutic efficacy. It is not clear whether these substances penetrate the blood-brain barrier, the issues of their bioavailability and the fact that binding to molecules in general will have a positive effect are not clear. Are there other known ligands or drugs for these proteins? COX6B1 is a structural subunit of the end enzyme of the mitochondrial respiratory chain. What is the biological significance of the binding of the identified substances to it? This binding can potentially have a negative meaning. It is recommended to revise this section and conduct a more critical analysis. The study should not mechanically list the data, but should have a critical biological or pharmacological analysis.

Response:

We fully agree with the reviewer that this section required a more critical and scientifically cautious framing. We have substantially revised the manuscript to transform this section from a declarative report into a hypothesis-generating exploration, with explicit discussion of the numerous caveats.

Our key revisions are as follows:

  1. Re-framing the Section's Intent: We have revised the title of Section 3.7 to "Knowledge-Based Screening... and In Silico Assessment" (from "...and Molecular Docking Validation") and have rephrased the text to clarify that this analysis is for "generating pharmacological hypotheses" rather than assessing "therapeutic potential." (Page 28, Line 948-949)
  2. Critical Analysis of Molecular Docking Results: In Section 3.7, we now explicitly state that a favorable docking score indicates structural plausibility but "does not elucidate the functional consequence of the interaction." We directly address the reviewer's point about COX6B1 by stating that binding could lead to beneficial stabilization, allosteric modulation, or even "detrimental inhibition of the respiratory chain." This acknowledges the ambiguity and critical need for experimental follow-up. (Page 29, Line 982-990)
  3. Addressing Pharmacological Hurdles in the Discussion: We have rewritten the corresponding paragraph in the Discussion (Section 4) to emphasize the major unaddressed hurdles. We now clearly state that our analysis did not evaluate "key pharmacokinetic properties, such as the ability of these compounds to penetrate the blood-brain barrier or their bioavailability." (Page 31, Line 1088-1098)
  4. Acknowledging Scope Limitations: In the Limitations section (Section 4), we have added statements to formally acknowledge that our screening "did not include a systematic search for other known... ligands" and reiterated that pharmacokinetic properties were not assessed. This transparently defines the boundaries of our computational work. (Page 32, Line 1142-1153)

We believe these comprehensive revisions have fundamentally improved this part of the manuscript. The analysis is now presented with appropriate scientific skepticism, directly addressing the critical biological and pharmacological questions raised by the reviewer. We thank the reviewer for pushing us to elevate the quality of this analysis significantly.

Reviewer 3 Report

Comments and Suggestions for Authors

Dear authors. Good work has been done on relevant topics. The data landscape is quite extensive and harmoniously structured. I don't see any flaws. The article can be recommended for publication after an editorial review.

Some minor comments are attached.

Author Response

Reviewer 3

  1. A simple ref is enough. IDs are not required

Response:

We thank the reviewer for this helpful suggestion regarding citation formatting. We agree that a simplified reference format is cleaner and more reader-friendly.

Accordingly, we have revised Table 2 as requested. We have removed the PubMed IDs (PMIDs) from the "Publications" column and now provide a standard citation format (e.g., Piras IS, et al., J Alzheimers Dis, 2019), as can be seen in the updated table. This change improves the table's readability and consistency with standard academic formatting.

  1. Give references to the software

Response:

We thank the reviewer for this important suggestion to improve the formal citation of our software environments. We agree that providing references for core tools like R and Python is crucial for reproducibility and proper attribution.

Accordingly, we have updated the "Statistical analysis" section (2.13) to include appropriate citations for the R/Bioconductor project (Huber et al., Nat Methods, 2015) and for Python in scientific computing (Millman & Aivazis, Comput Sci Eng, 2011). This ensures that these foundational software environments are properly referenced in our manuscript.

  1. explain the choice of test objects.

Response:

We thank the reviewer for these crucial questions regarding the methodology of our network analysis. We acknowledge that the selection criteria for both the gene targets and the corresponding herbs were not sufficiently detailed. We have revised the legend for Figure 6(C) to provide a complete and transparent explanation of our data-driven approach.

Our response addresses both points:

  1. Choice of Gene Targets ("Test Objects"): As explained in our previous response, we have clarified that the six gene targets are the elite candidates that emerged from our multi-stage pipeline, selected based on a convergence of genetic, functional, and predictive evidence.
  2. Selection of Herbs: This is a key point, and we have now made our methodology explicit in the figure legend. The process was gene-centric, not herb-centric. We did not start with a list of herbs to test. Instead, the workflow was as follows:
    • First, we finalized our six high-confidence gene targets.
    • Second, we performed a comprehensive query against the integrated TCM database to identify all documented bioactive molecules known to interact with these six specific gene targets.
    • Finally, the herbs displayed in the Sankey diagram are the direct output of this query. They are simply the documented botanical sources of the identified molecules.

Crucially, this means there was no pre-selection or "cherry-picking" of herbs. The inclusion of an herb in the figure is based entirely on whether it contains a molecule with existing pharmacological evidence linking it to one of our data-driven gene targets. This ensures the analysis is unbiased and directly tied to the primary findings of our study. We believe this clarification significantly improves the transparency and rigor of our work.

  1. Divide it into separate figures. It's better for perception.

Response: We thank the reviewer for this suggestion aimed at improving the perceptual clarity of our figures. We have carefully considered this point and would like to respectfully explain the scientific rationale for maintaining the current composite format for these specific figures. We believe that while the figures are information-dense, their current structure is integral to conveying the key comparative insights of our study.

  • For Figure 2C (Causal Effect Matrix): The primary goal of this figure is to provide a direct visual demonstration of the asymmetrical causal relationship between neurocognitive traits and sleep. The matrix format allows the reader to immediately perceive the high density of significant associations in one direction versus the sparseness in the other. We believe that splitting this into separate plots would dismantle this central, comparative message and obscure the study's key finding regarding directionality.
  • For Figure 3C (Faceted Manhattan Plots): This figure is intentionally designed as a comprehensive landscape to facilitate direct comparison of SMR results across different traits and multi-omic layers. The faceted design allows for the rapid identification of crucial patterns, such as cross-trait pleiotropy (e.g., a locus appearing in multiple trait panels) and the relative contribution of different molecular QTLs (e.g., pQTLs vs. eQTLs) to disease risk. Separating these into individual plots would result in fragmented information and prevent the reader from making these essential horizontal and vertical comparisons.
  • For Figure 5B (Top Significant Expression Changes): This panel is designed to present a cohesive summary of the complex, cell-type-specific dysregulation patterns for our final set of prioritized genes. By presenting these examples together, we allow for direct comparison of the regulatory logic between these key genes, highlighting central themes such as the opposing expression patterns in neurons versus glia. We feel that separating them would disrupt this narrative and weaken the evidence for a coordinated dysregulation of this core gene signature.

In summary, the composite nature of these figures is a deliberate design choice to emphasize the comparative and integrative aspects of our analyses. We hope this explanation clarifies our rationale for retaining the current format, which we believe most effectively communicates the scientific story of the manuscript.

Round 2

Reviewer 1 Report

Comments and Suggestions for Authors

The authors are basically able to modify the manuscript according to my comments. In this case, they can consider receiving the manuscript.

Reviewer 2 Report

Comments and Suggestions for Authors

The authors answered my questions and made corrections to the text of the article, properly correcting my concerns.